# Suppression of cGAS- and RIG-I-mediated innate immune signaling by Epstein-Barr virus deubiquitinase BPLF1

**Wai-Yin Lui[1], Aradhana Bharti[2], Nok-Hei Mickey Wong[1¤a], Sonia Jangra[2¤b], Michael G. Botelho[2], Kit-San Yuen[1,3]\*, Dong-Yan Jin[1]\***

**1** School of Biomedical Sciences, the University of Hong Kong, Pokfulam, Hong Kong, **2** Faculty of Dentistry, the University of Hong Kong, Sai Yin Pun, Hong Kong, **3** School of Nursing, Tung Wah College, Kowloon, Hong Kong

¤a Current address: Department of Haematological Medicine, Division of Cancer Studies, Leukemia and Stem Cell Biology Team, King's College London, London, United Kingdom
¤b Current address: Department of Microbiology, Icahn School of Medicine at Mount Sinai, New York, New York, United States of America
\* samyuen@hku.hk (K-SY); dyjin@hku.hk (D-YJ)

**Data Availability Statement:** All relevant data are within the manuscript and its Supporting information files.

## Abstract

Epstein-Barr virus (EBV) has developed effective strategies to evade host innate immune responses. Here we reported on mitigation of type I interferon (IFN) production by EBV deubiquitinase (DUB) BPLF1 through cGAS-STING and RIG-I-MAVS pathways. The two naturally occurring forms of BPLF1 exerted potent suppressive effect on cGAS-STING-, RIG-I- and TBK1-induced IFN production. The observed suppression was reversed when DUB domain of BPLF1 was rendered catalytically inactive. The DUB activity of BPLF1 also facilitated EBV infection by counteracting cGAS-STING- and TBK1-mediated antiviral defense. BPLF1 associated with STING to act as an effective DUB targeting its K63-, K48- and K27-linked ubiquitin moieties. BPLF1 also catalyzed removal of K63- and K48-linked ubiquitin chains on TBK1 kinase. The DUB activity of BPLF1 was required for its suppression of TBK1-induced IRF3 dimerization. Importantly, in cells stably carrying EBV genome that encodes a catalytically inactive BPLF1, the virus failed to suppress type I IFN production upon activation of cGAS and STING. This study demonstrated IFN antagonism of BPLF1 mediated through DUB-dependent deubiquitination of STING and TBK1 leading to suppression of cGAS-STING and RIG-I-MAVS signaling.

## Author summary

EBV is an oncogenic virus causing Burkitt lymphoma and nasopharyngeal cancer. Elucidating how the virus evades human innate immune responses can provide a better understanding of how EBV causes cancer. In this study we have screened all EBV proteins to identify BPLF1 as a virulence factor that contributes to innate immune evasion. BPLF1 fulfills this function by removing the polyubiquitin marks on critical effector proteins including STING and TBK1. We have also established cell lines stably carrying a

**Funding:** This study was supported by grant 18170942 to D.-Y.J. from Hong Kong Health and Medical Research Fund and grant 17127019 to D.-Y. J. from Hong Kong Research Grants Council. The funders had no role in study design, data collection and analysis, decision to publish, or preparation of the manuscript.

**Competing interests:** The authors have declared that no competing interests exist.

recombinant EBV that produces a catalytically inactive BPLF1. The recombinant virus has lost the ability to suppress antiviral responses induced by cGAS and STING in these cells. Our results provide mechanistic insight on the suppression of type I interferon production by BPLF1. Our study might also reveal new knowledge of EBV pathogenesis and new strategies for antiviral development.

## Introduction

Epstein-Barr virus (EBV), a γ-herpesvirus, is a ubiquitous DNA virus that persistently infects more than 90% of the human populations [1]. This virus is the first oncogenic virus identified and EBV infection is etiologically associated with 2% of human malignancies, including nasopharyngeal carcinoma particularly prevalent in Hong Kong and nearby regions of China [2]. Primary infection of EBV during adolescence may result in the development of infectious mononucleosis [3], but contracting the virus at an earlier age is mostly asymptomatic. Following the infection at the epithelial cells, the virus would undergo lytic replication to infect the lymphoid tissues and eventually migrate to the resting memory B cells to establish long-term persistent infection [4]. During latent infection, the virus might exhibit EBNA-1-only program and keeping viral gene expression to the minimum is an effective immune evasion strategy [5,6].

Innate immunity serves as the first line of defense for vertebrates to protect against pathogens [7,8]. Type I interferons (IFNs) are a group of cytokines produced by most cells during pathogen invasion [9]. Upon EBV primary infection, multiple cytosolic DNA sensors and RNA sensors are activated to mount appropriate immune responses [10,11]. The presence of cytosolic DNA such as EBV genome [12] would trigger DNA sensor cyclic GMP-AMP synthase (cGAS) to induce type I IFN response via the cGAS-STING-TBK1 axis [13,14], while the prevalence of cytosolic RNA such as EBV-encoded small RNAs (EBERs), EBV-encoded circular RNAs and cellular 5S rRNA would activate RNA sensor retinoic acid-inducible gene I (RIG-I) to activate the production of type I IFNs through the RIG-I-MAVS-TBK1 axis [14–17]. The activated TBK1 will then activate the downstream transcription factor IRF3 to produce IFN-β.

Ubiquitination is an important post-translational modification (PTM) that regulates the biological functions of proteins. Ubiquitin is an 8kDa peptide with seven lysine residues (K6, K11, K27, K29, K33, K48 and K63) [18]. The two most common types of ubiquitination are the K48- and K63-linked ubiquitination. Whereas K48-ubiquitination usually leads to proteasome degradation of the target protein, K63-ubiquitination mostly signals for non-proteolytic regulatory roles such as kinase activation and pathway switch. In the cGAS-STING pathway, multiple effectors have been ubiquitinated. In particular, K48-linked ubiquitination of adaptor protein STING results in proteasome degradation, while K63-linked ubiquitination activates STING for type I IFN production [19]. K27-linked ubiquitination on STING serves as a docking site for TBK1 binding and K11-linked ubiquitination confers protective effect against degradation [20,21]. Likewise, K48-linked ubiquitination of TBK1 targets the protein for degradation and K63-linked ubiquitination of TBK1 leads to enhancement of type I IFN production [22,23]. RIG-I and MAVS in the RIG-I-MAVS pathway are also subjected to ubiquitination, with K48-linked ubiquitination leading to degradation and K63-linked ubiquitination resulting in protein activation [24–26].

To evade host innate and adaptive immune responses directed against EBV proteins and RNAs [14,15,27,28], EBV has developed multiple strategies to facilitate primary infection and

establishment of lifelong persistence [11,29]. Many EBV-encoded proteins and RNAs have been found to carry immunosuppressive abilities that can prohibit immune responses during primary infection and reactivation [11]. Cellular antiviral signaling pathways such as type I IFN, NF-κB and Toll-like Receptor (TLR) are effectively suppressed by EBV [29,30].

BPLF1 is a 350kDa large tegument protein made up of 3149 amino acids [31]. The large tegument protein is normally expressed during the lytic phase of EBV infection and is packaged into the infectious virion [32]. The protein has previously been reported to have deneddylase and deubiquitinase (DUB) activities, both of which are mediated by the same conserved cysteine protease domain located at the N-terminus [33]. There are two naturally occurring forms of BPLF1, the full-length BPLF1 and the small fragment of BPLF1 composed of the first 325 amino acids (BPLF1 325). Both versions are expressed during productive EBV infection. The full- length BPLF1 can be found in infectious EBV virion, while BPLF1 325 appears in infected cells 24–48 hours after productive EBV infection [31]. BPLF1 325 consists of the full DUB domain and is widely employed in functional analysis of BPLF1 DUB. BPLF1 knock-out and knock-down have shown remarkable decrease in EBV infectivity in epithelial cells and B cells, suggesting the paramount importance of BPLF1 to viral immune evasion and replication [34]. Mechanistically, BPLF1 has been shown to suppress Toll-like receptor and NF-κB signaling [31,35]. It also targets TRIM25, TRAF6 and SQSTM1/p62 to inhibit innate immune response and selective autophagy [35–37]. Other known targets of BPLF1 include PCNA, Rad18, translesion polymerase η and TOP2 involved in cellular DNA repair and translesion synthesis [38–41], as well as EBV ribonucleotide reductase [42], which inhibits APOBEC3 to preserve viral genome integrity [43].

In this study, BPLF1 was found to contribute to EBV innate immune evasion by deubiquitinating adaptor proteins STING and TBK1, thereby inhibiting cGAS-STING- and RIG-I-MAVS-induced type I IFN production. BPLF1 was found to remove all types of ubiquitin moieties on STING and TBK1 molecules, prohibiting the activities of these proteins. We have also established cell lines stably carrying recombinant EBV genome with catalytically inactive BPLF1. The BPLF1-inactive virus was incapable of suppressing cGAS-STING-mediated antiviral response. Our work delineated a new mechanism by which EBV BPLF1 facilitates viral evasion from host innate immune surveillance.

## Results

### EBV expression library screening for IFN-β antagonists

In search of EBV-encoded IFN-β antagonists, an EBV expression library screen was performed in HEK293 cells using IFNB-Luc reporter, a *Firefly* luciferase construct driven by IFN-β promoter. Since DNA sensing is most relevant to EBV infection [11,12], cGAS and STING were overexpressed to induce IFNB-Luc activity. Among 53 EBV proteins, most were neutral on cGAS-STING-induced activation of IFNB-Luc. Potent suppressive effects were observed for viral proteins BPLF1, Zta, Rta, BMRF1, BLRF2, BSRF1, BBLF2 and BGLF2, whereas BERF1, BaRF1 and BLLF3 exhibited further stimulatory effects on cGAS and STING (Fig 1A). Since this was the first indication that some EBV proteins might be potent suppressors of cGAS-STING signaling, it is of particularly great interest to investigate how they would antagonize the activity of cGAS and STING. In view of its potent suppressive effect, BPLF1 was chosen for our further analysis in this study.

To validate the effects of BPLF1 on cGAS-STING signaling, BPLF1 325 (referred to as BPLF1 hereafter for simplicity) was ectopically expressed in cGAS-deficient HEK293 cells together with cGAS and STING, side by side with RIG-I N, which is a dominant active form of RIG-I, and TBK1. The expression of BPLF1 potently suppressed the cGAS-STING-induced

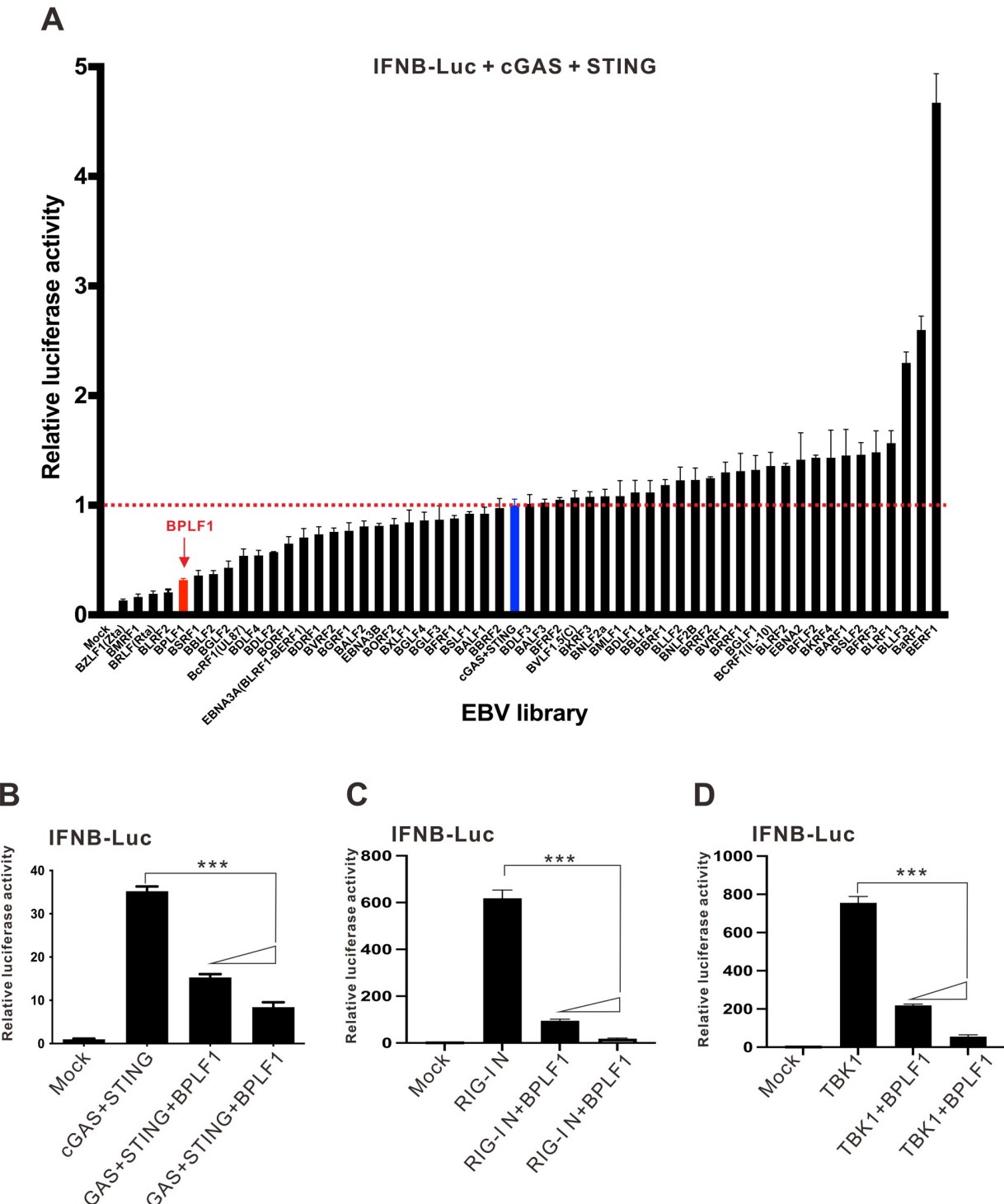

**Fig 1. BPLF1 is an IFN antagonist.** (A) Functional screening for IFN-β antagonists. HEK293 cells were transfected with a *Firefly* luciferase reporter plasmid driven by IFN-β-promoter (IFNB-Luc), a control *Renilla* luciferase reporter plasmid driven by thymidine kinase promoter (TK-Luc) serving to normalize for transfection efficiency, the plasmids for EBV protein expression library, together with cGAS and STING expression plasmids. Cells were harvested 24 hours post-transfection for dual-luciferase assay. The mean values of three biological replicates (n = 3) were represented by the bars and their respective standard deviations were depicted as the error bars. (B-D) BPLF1 inhibits cGAS-STING-, RIG-I- and TBK1-induced IFNβ-promoter activation. HEK293 cells were transfected with IFNB-Luc, TK-Luc, and increasing doses of BPLF1 expression plasmid. IFNB-Luc reporter expression was stimulated with expression plasmids for cGAS + STING (B), RIG-I N (C) and TBK1 (D). Cells were harvested 24 hours post-transfection for dual-luciferase assay. The mean values of three biological replicates (n = 3)

were represented by the bars and their respective standard deviations were depicted as the error bars. The statistical significance for the difference between the indicated groups was analyzed using two-tailed Student's t-test for paired samples and the ranges of the $p$ values were indicated (*: $p < 0.05$; **: $p < 0.01$; and ***: $p < 0.001$).

activation of IFNB-Luc activity (Fig 1B). Similar inhibition of IFNB-Luc activity was also observed when BPLF1 was co-expressed with RIG-I N or TBK1 in HEK293 cells (Fig 1C and 1D). Thus, BPLF1 might act as a potent suppressor of both DNA sensing and RNA sensing pathways.

## IFN antagonism of BPLF1

We next sought to shed more light on the action point of BPLF1 in the suppression of IFN-β production. Whereas BPLF1 blunted the ability of cGAS and STING to induce IFN-β gene transcription (Fig 2A), it had no effect on the activity of transcription factor IRF3 to activate IFNB-Luc activity (Fig 2B). A more detailed analysis indicated that two doses of BPLF1 had no influence on the activity of IRF3-5D, which is the dominant active form of IRF3 with 5 phosphomimetic substitutions S396D, S398D, S402D, T404D and S405D [44], to activate IFNB-Luc reporter expression (Fig 2C). To verify the specificity of the inhibitory action of BPLF1, the impact of BPLF1 on luciferase reporter expression driven by SV40 promoter was assessed. No remarkable difference in SV40 promoter activity can be observed when BPLF1 was expressed (Fig 2D), demonstrating that the observed suppression of IFNB-Luc activity was unlikely mediated through non-specific transcriptional suppression. Collectively, our results suggested that BPLF1 inhibits cGAS-STING- and RIG-I-induced activation of IFN-β production specifically at a step upstream of IRF3.

Due to its large size, full-length BPLF1 was difficult to clone and express from a plasmid. To rectify this, CRISPR-a technology [45] was employed to induce the expression of full-length BPLF1 endogenously from the EBV genome using guide RNA (gRNA) (Fig 2E and 2F). The catalytically inactivated "dead" Cas9 (dCas9) protein linked to four tandem copies of herpes simplex virus 1 (HSV-1) trans-activator VP16, also known as VP64, was targeted to the desired DNA sequence with the gRNA (Fig 2E). Several transcription factors would be recruited to the complex to enhance the endogenous expression of full-length BPLF1 from the EBV genome in HEK293-M81 cells constitutively carrying EBV M81 strain [46]. When HEK293-M81 cells were co-transfected with gRNAs and lenti-MPHv2 expressing the relevant transcription factors, significant increase in endogenous BPLF1 transcript was observed, indicating that full-length BPLF1 can be induced by CRISPR-a (Fig 2G) without inducing lytic reactivation (S1A Fig).

The effect of full-length BPLF1 on cGAS-STING-, RIG-I N-induced type I IFN production was then assessed using luciferase reporter assays. When the expression of full-length BPLF1 was induced in HEK293-M81 cells, significant suppression was observed for cGAS-STING-, RIG-I N-, TBK1-induced activation of IFNB-Luc and IRF3-Luc activity, but not for IRF3-5D-indcued activation of IRF3-Luc activity (Fig 2H and S1B Fig). The inhibitory effect of full-length BPLF1 on TBK1-induced IFNB-Luc activity was more robust when higher dose of gRNA was transfected (Fig 2I). Thus, induced expression of full-length BPLF1 also exhibited an inhibitory effect on IFN-β production.

BPLF1 is known to affect NF-κB pathway at multiple levels [35]. Since the IFN-β promoter in IFNB-Luc is controlled by three transcription factors IRF3, AP-1 and NF-κB [47], it would be necessary to clarify whether the observed suppressive effect of BPLF1 on IFNB-Luc may be ascribed to its action on the NF-κB pathway, rather than on the IRF3 pathway. Another

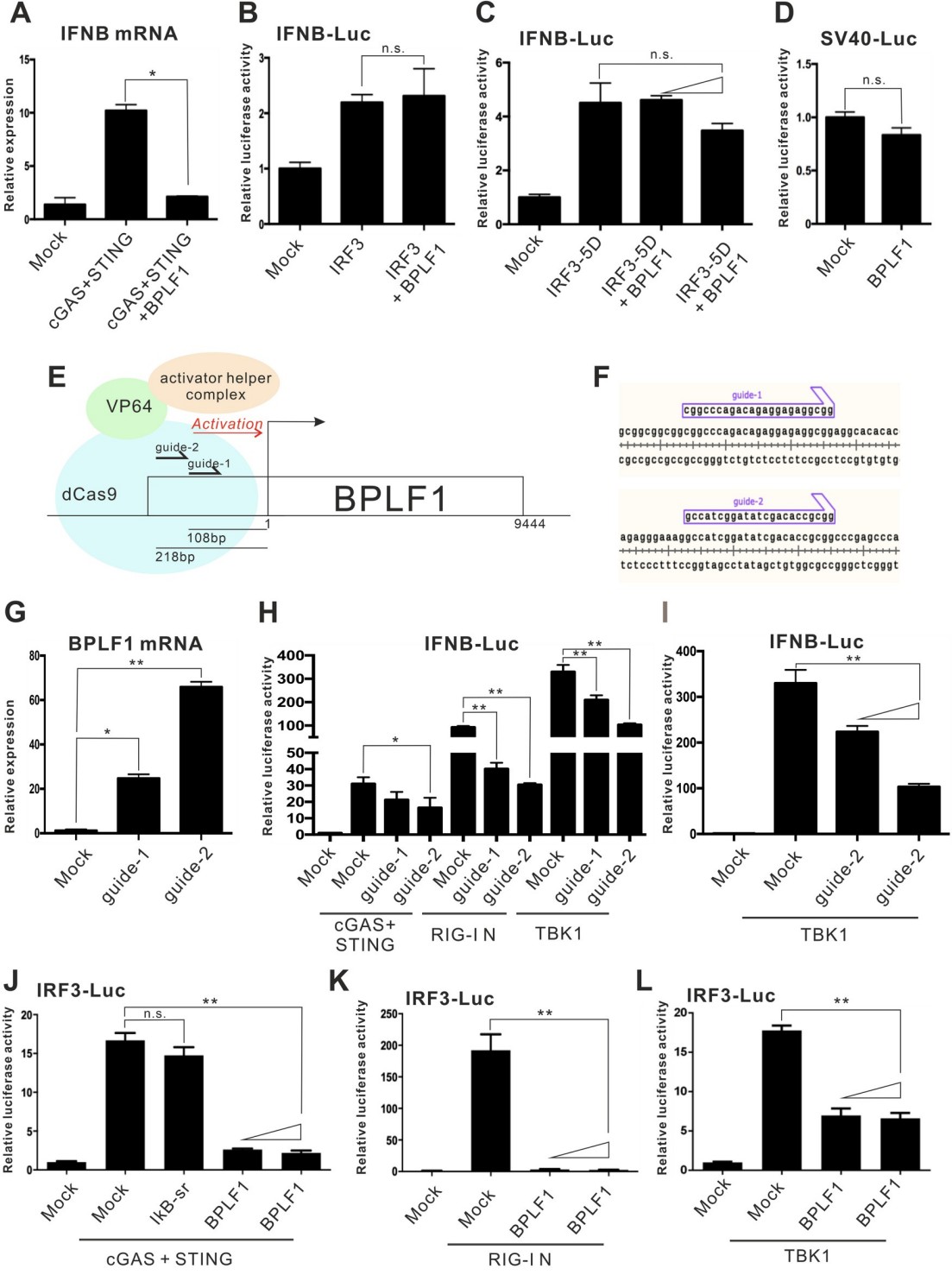

**Fig 2. IFN antagonism of BPLF1.** (A-D) Specific inhibition of cGAS-STING activity by BPLF1. HEK293 cells were transfected with IFNB-Luc, TK-Luc, either one or two doses of BPLF1 expression plasmid, and expression plasmids for cGAS + STING (A), IRF3 (B) or IRF3-5D (C). Alternatively, a *Firefly* luciferase reporter driven by the SV40-promoter (SV40-Luc) and TK-Luc promoter were co-transfected with BPLF1 expression plasmid into HEK293 cells (D). Cells were harvested 24 hours post-transfection for dual-luciferase assay. (E) A schematic diagram depicting the targeting regions of the two gRNAs (guide-1 and guide-2) on the EBV genome was presented together with dCas9, four tandem copies of HSV-1 VP16 molecules (VP64) and the activator helper complex. (F) The cDNA sequences of the two gRNAs. (G) Induction of BPLF1 expression by CRISPR-a. HEK293M81 cells were transfected with lentiMPHv2 and lentiSAMv2 carrying guide-1 and guide-2. The total cellular RNA was

harvested 48 hours post-transfection and the BPLF1 mRNA was measured by RT-qPCR. The BPLF1 mRNA expression levels were normalized to those of endogenous GAPDH transcript. (H) IFN antagonism of BPLF1 induced to express from EBV genome by CRISPR-a. HEK293M81 cells were transfected with IFNB-Luc, TK-Luc, guide-1 or guide-2 plasmids and expression plasmids for cGAS + STING, RIG-I N and TBK1. (I) Dose-dependent IFN antagonism of BPLF1 expressed from EBV genome. HEK293M81 cells were transfected with IFNB-Luc, TK-Luc, increasing doses of guide-2 plasmid, and TBK1 expression plasmid for dual luciferase assays. (J-L) BPLF1 inhibits cGAS-STING-, RIG-I- and TBK1-induced activation of IRF3-binding elements. HEK293 cells were transfected with a *Firefly* luciferase reporter plasmid driven by IRF3-Luc, TK-Luc, increasing doses of BPLF1 expression plasmid, and expression plasmids for cGAS + STING (J), RIG-I N (K) or TBK1 (L). IκB super-repressor was also over-expressed in (J). Cells were harvested 24 hours post-transfection for dual-luciferase assay. The mean values of three biological replicates (n = 3) were represented by the bars and their respective standard deviations were depicted as the error bars. The statistical significance among selected samples was analyzed using two-tailed Student's t-test for paired samples and the ranges of the *p* values were indicated (*: $p < 0.05$; **: $p < 0.01$; and n.s.: not significant or $p > 0.05$).

luciferase reporter, IRF3-Luc, which consists of 5 IRF3-binding elements only [44,48], was then employed in this study to solely reflect the effect of BPLF1 on IRF3-dependent transcription.

To rule out the possibility that NF-κB might activate IRF3-Luc activity in connection to crosstalk between NF-κB and cGAS-STING pathways [49,50], a NF-κB suppressor known as IκB super repressor (IκB-sr) [51], was co-expressed with cGAS and STING in HEK293 cells. The activity of IκB-sr to suppress p65-induced NF-κB activity was verified in S1C Fig. If the IRF3-Luc was responsive to NF-κB pathway, the presence of IκB-sr would remarkably suppress IRF3-Luc activity. However, here we failed to observe any significant difference in IRF3-Luc activity when IκB-sr was present (Fig 2J), suggesting that IRF3-Luc is not responsive to NF-κB signaling.

BPLF1 was then ectopically expressed in HEK293 cells co-expressing cGAS and STING, RIG-I N, or TBK1. Introduction of BPLF1 into these cells prominently suppressed the induced IRF3-Luc activity (Fig 2J–2L). These results indicated that BPLF1 suppresses cGAS-STING-, RIG-I N, TBK1-induced activation of IRF3.

To further characterize the ability of full-length BPLF1 to suppress cGAS-STING signaling in response to ligands such as dsDNA, we treated HEK293M81 cells overexpressing cGAS and STING with ISD90 and induced the expression of full-length BPLF1 for an IRF3-Luc luciferase assay. Upon ISD90 treatment, more robust IRF3-Luc activity was observed, but the induction of full-length BPLF1 expression mitigated the ISD90-induced IRF3-Luc activity in a dose-dependent manner (S2 Fig).

## DUB is required for IFN antagonism of BPLF1

BPLF1 has demonstrated potent DUB activity in relation to its suppression of NF-κB signaling [35]. To investigate if the DUB activity of BPLF1 is required for its inhibition of IRF3 signaling, a previously described DUB-dead mutant C61A (referred to as C61A hereafter) [31] was constructed. The DUB domain of BPLF1 was inactivated by replacing the cysteine residue at position 61 to alanine. C61A can be ectopically expressed in HEK293 cells (Fig 3A). C61A was employed in multiple assays to determine the requirement of the DUB activity for IFN antagonism.

When BPLF1 was replaced by C61A, the suppression on cGAS-STING-induced IRF3-Luc activity was less robust (Fig 3B). Similar results were observed in RIG-I N- and TBK1-overexpressing cells and the suppression of IRF3-Luc activity by BPLF1 was reversed when C61A was expressed (Fig 3C and 3D). The essentiality of the DUB domain for IFN-β induction was also assessed by RT-qPCR. When BPLF1 was replace by C61A, the IFN-β transcript level was restored in the cGAS-STING-overexpressing HEK293 cells (Fig 3E). This indicated that BPLF1 antagonizes IFN-β gene transcription via its DUB domain.

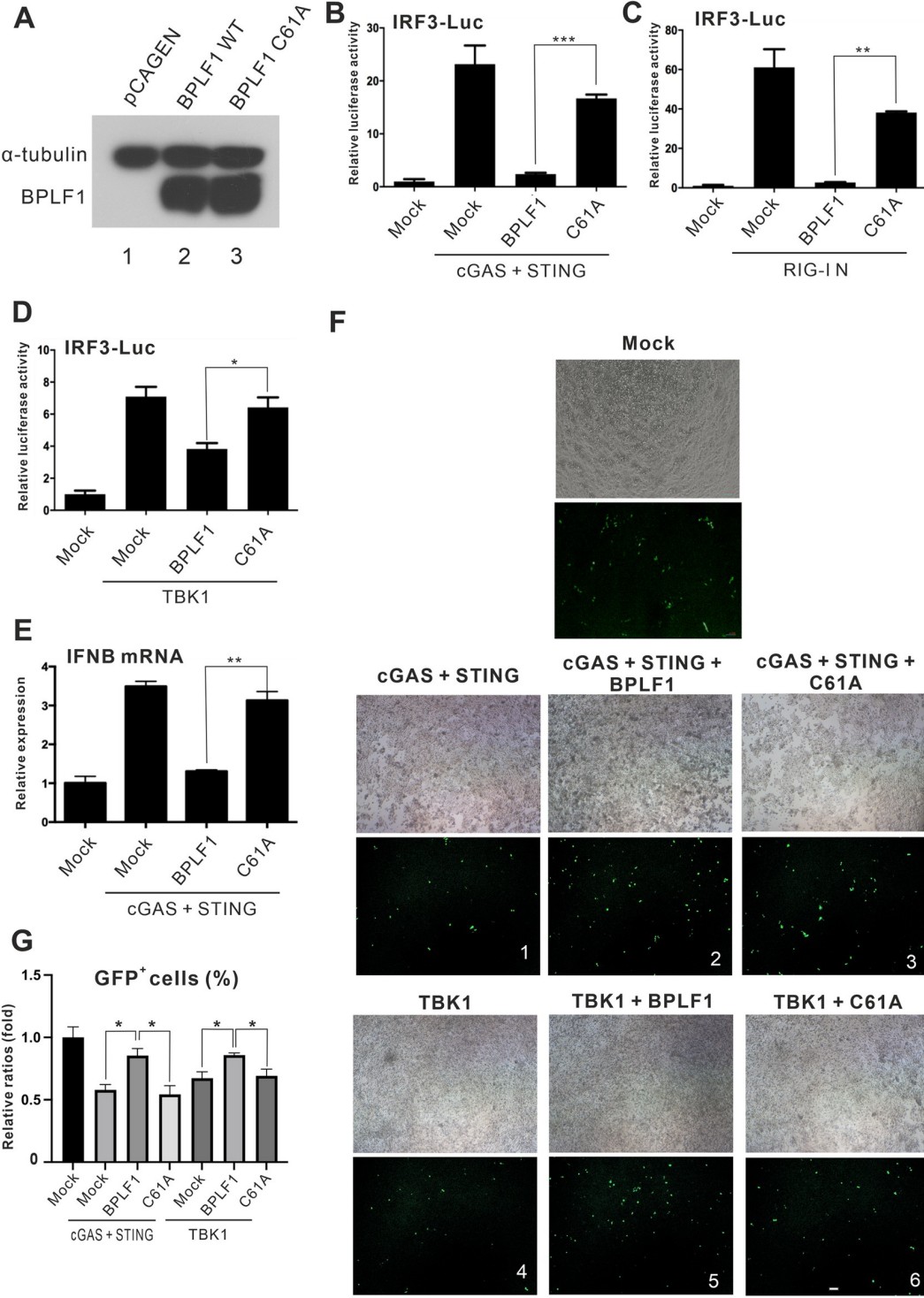

**Fig 3. Requirement of deubiquitinase domain (DUB) for IFN-inhibitory effect of BPLF1.** (A) Expression of BPLF1 C61A mutant. BPLF1 WT and C61A were transiently expressed in HEK293 cells. Endogenous α-tubulin was used for normalization. (B-D) IRF3 activation by C61A mutant. The indicated combinations of BPLF1 expression plasmid, mutant C61A expression construct as well as IRF3-Luc and TK-Luc reporter plasmids were transfected into HEK293 cells together with cGAS + STING (B), RIG-I N (C) or TBK1 (D) expression plasmids. Cells were harvested 24 hours post-transfection for dual-luciferase assay. (E) Effect of C61A on IFN-β induction. HEK293 cells were transfected with cGAS, STING, BPLF1 and mutant C61A expression plasmids. The total cellular RNA was harvested 48 hours post-transfection and the IFN-β transcripts expression levels were measured by RT-qPCR. The IFNβ transcript expression levels were normalized to those of endogenous

GAPDH transcript. The mean values of three biological replicates (n = 3) were represented by the bars and their respective standard deviations were depicted as the error bars. (F, G) The DUB activity of BPLF1 promotes EBV infection. BPLF1 and C61A expression plasmids were transfected into HEK293 cells together with cGAS + STING expression plasmids (panels 1–3) and TBK1 expression plasmid (panels 4–6). After 24 hours post- transfection, cells were infected with 1 m.o.i. of freshly prepared EBV M81. Microscopic images were captured (F) and GFP signals were analyzed by flow cytometry 48 hours post-infection (G). The percentages of GFP⁺ cells were normalized to the mock infection group. The mean values of three biological replicates (n = 3) were represented by the bars and their respective standard deviations were depicted as the error bars. The statistical significance among selected samples was analyzed using two-tailed Student's t-test for paired samples and the ranges of the $p$ values were indicated (*: $p < 0.05$; and **: $p < 0.01$; ***: $p < 0.001$).

To assess the impact of BPLF1 and its DUB activity on EBV infection, different combinations of cGAS + STING, TBK1, BPLF1 and C61A were expressed in HEK293 cells, which were subsequently infected with freshly prepared GFP-marked EBV M81 virus 24 hours after transfection. Successful EBV infection would produce GFP signal and the percentages of GFP⁺ cells were then measured by flow cytometry. When BPLF1 was co-expressed with cGAS + STING or TBK1, more GFP⁺ cells (Fig 3F, panels 2 and 5) and significantly higher percentages of GFP⁺ cells were being detected (Fig 3G). In contrast, no noticeable increase in the percentage of GFP⁺ cells was observed when C61A was expressed (Fig 3F, panels 3 and 6 and Fig 3G). These results illustrated that the DUB activity of BPLF1 facilitates primary EBV infection upon activation of DNA sensing and RNA sensing.

## BPLF1 perturbs STING ubiquitination

Adaptor proteins STING and TBK1 are critically regulated by ubiquitination [52]. The different types of ubiquitination on these adaptor proteins lead to their activation, degradation or signal transduction [19–23,52]. With this in mind, we asked whether ubiquitination on effector proteins STING and TBK1 might be affected by the DUB activity of BPLF1.

The ubiquitination patterns of STING were examined in the presence of BPLF1. BPLF1 was co-expressed with cGAS, STING, HA-tagged wild-type ubiquitin (HA-WT-Ub) and HA-tagged lysine-free ubiquitin (HA-Ub-K0) in HEK293 cells. Flag-tagged STING was immunoprecipitated with anti-Flag antibody. The ubiquitination of STING was seen as a ladder of bands extending upwards starting from the position of unmodified STING protein since ubiquitin moieties of different lengths were attached to the protein (Fig 4A, lane 2). The ubiquitin ladder of STING can be observed in cGAS-STING-overexpressing cells. When BPLF1 was introduced into the cells, the ubiquitination of STING was diminished by more than 5-fold, even if normalized to the steady-state level of STING (Fig 4A, lane 4). When BPLF1 was replaced by C61A, no remarkable decline in ubiquitination was observed (Fig 4A, lane 6). HA-Ub-K0 that cannot be added to the polyubiquitin chain served as a negative control for the experiments. The results indicated that BPLF1 de-ubiquitinates STING, leading plausibly to mitigation of cGAS-STING-induced IFN-β production. Thus, BPLF1 was influential on the PTM of STING.

To further characterize de-ubiquitination of STING by BPLF1, K63-, K48- and K27-linked ubiquitin molecules were expressed in HEK293T cells together with cGAS and STING. The overexpression of these ubiquitin moieties would lead to predominance of the specific type of ubiquitin linkage in the cells. Meanwhile, K63R- and K48R-linked ubiquitin molecules were also expressed. K63R-linked ubiquitin supports all types of polyubiquitination other than K63-linked ubiquitination, while K48R-linked ubiquitin assembles all types of polyubiquitin chain except the K48-linked version. By overexpressing these ubiquitin mutants, the effect of BPLF1 on each type of ubiquitination linkage can be elucidated.

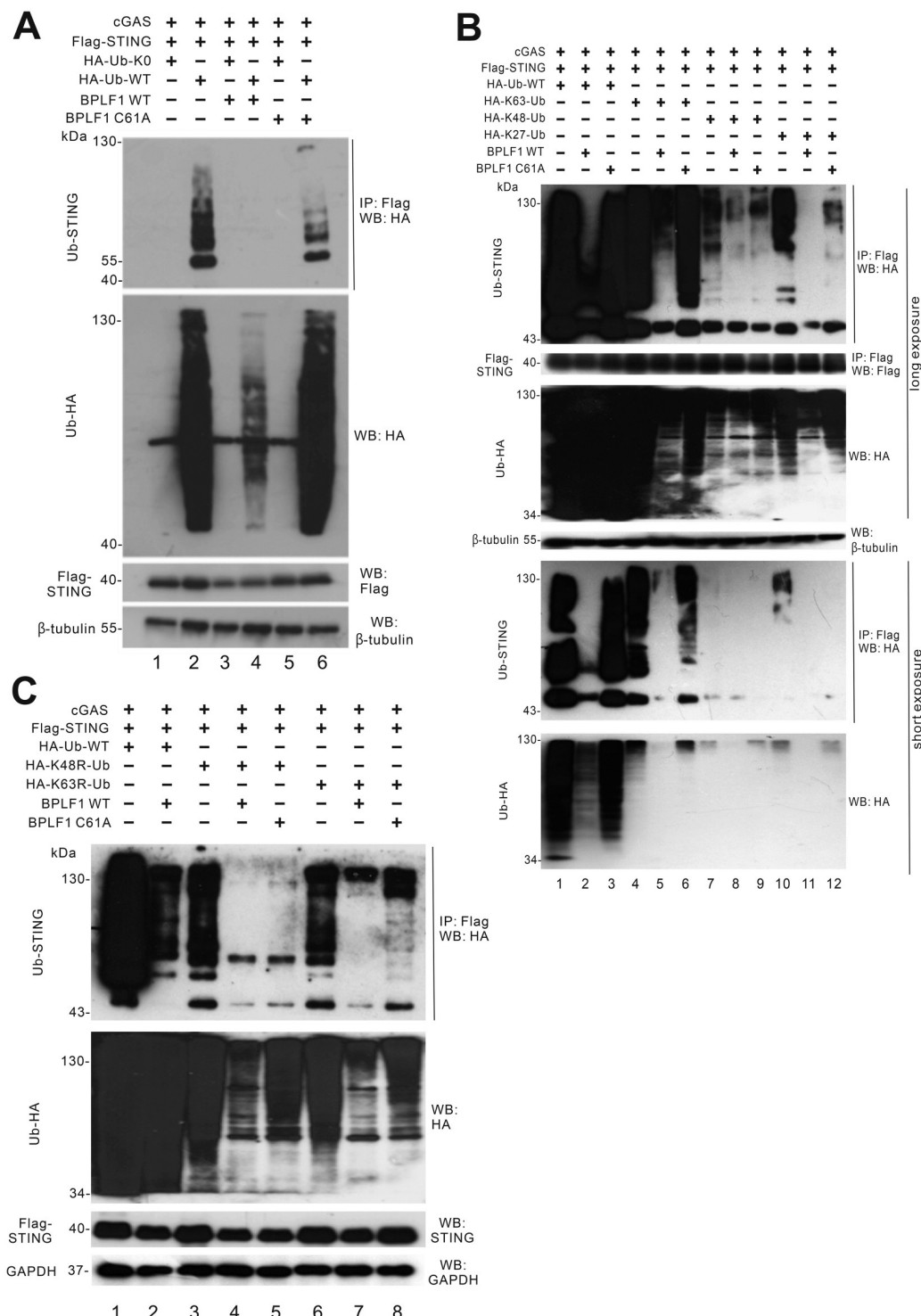

**Fig 4. BPLF1 perturbs STING ubiquitination.** (A) Influence of BPLF1 and C61A on STING ubiquitination. BPLF1, C61A, cGAS and STING expression plasmids were transfected into HEK293 cells together with HA-Ub-WT and HA-Ub-K0. (B) Influence of BPLF1 and C61A on STING ubiquitination of different types. BPLF1, C61A, cGAS and STING expression plasmids were transfected into HEK293T cells together with HA-Ub-WT and HA-K63-Ub, HA-K48-Ub and HA-K27-Ub. (C) Influence of BPLF1 and C61A on K48R- and K63R-linked ubiquitination of STING. BPLF1, C61A, cGAS and STING expression plasmids were transfected into HEK293T cells together with HA-Ub-WT, HA-K63R-Ub and HA-K48R-Ub. Cells were harvested 24 hours after transfection for co-immunoprecipitation. The Flag-tagged STING molecules were pulled down by anti-Flag antibodies. The bound fraction of the immunoprecipitates (IP) and the total

lysates (input) were analyzed by Western blotting (WB) with anti-HA, anti-Flag and anti-β-tubulin antibodies. Long exposure was conducted to visualize proteins in the upper panel and short exposure was conducted to visualize other proteins in the lower panel.

Ubiquitination ladders were observed when HA-K63-Ub, HA-K48-Ub or HA-K27-Ub were co-transfected with cGAS and STING into HEK293T cells (Fig 4B, lanes 4, 7 and 10). However, the addition of BPLF1 significantly perturbed the formation of all three types of ubiquitination ladders (Fig 4B, lanes 5, 8 and 11). When C61A was expressed instead, no remarkable decline in K63-linked ubiquitination was observed, whereas K48- and K27-linked ubiquitination demonstrated partial reduction (Fig 4B, lanes 6, 9 and 12). These results showed that BPLF1 deubiquitinates K63-, K48- and K27-linked polyubiquitin chains on STING. Since K63-, K48- and K27-linked ubiquitination respectively activates STING [19], triggers its proteasomal degradation, and recruits TBK1 for signal transduction [20], the disturbance of all three types of ubiquitination on STING would therefore suppress the activation of STING and prohibit STING signaling without altering its protein expression level. Similar observations were also made when HA-Ub-K63R and HA-Ub-K48R were overexpressed in HEK293T cells (Fig 4C). These results suggested that BPLF1 might alter all types of polyubiquitin chains on STING.

To better investigate the deubiquitinating capability of BPLF1 on endogenous STING, cells were treated with ISD90 and then immunoprecipitated to examine the ubiquitin moieties. After 4 hours of ISD90 treatment, ubiquitin moieties were visible on endogenous STING protein in the immunoprecipitates (S2B Fig). The ubiquitination level of STING was remarkably decreased when BPLF1 was expressed, but partially restored when BPLF1 was replaced by BPLF1-C61A. These results demonstrated the deubiquitinating ability of BPLF1 on endogenous STING.

## BPLF1 perturbs TBK1 ubiquitination

Considering that TBK1 is a common downstream effector for both cGAS-STING and RIG-I-MAVS pathways and it is pivotally regulated by ubiquitination [53], the ubiquitination patterns on TBK1 were also examined. The ubiquitination ladders can be observed when wild type ubiquitin was expressed in HEK293T cells together with TBK1 (Fig 5A, lane 1). The presence of BPLF1 has led to remarkable decline in ubiquitination on TBK1 (Fig 5A, lane 2). When BPLF1 was substituted with C61A, restoration of the ubiquitination ladder of TBK1 was seen (Fig 5A, lane 3). These results showed that BPLF1 also de-ubiquitinates TBK1 with its DUB domain, leading to inactivation of the kinase.

The K63- and K48-linked ubiquitination patterns on TBK1 were also examined using HA-Ub-K63 and HA-Ub-K48 mutants. When HA-Ub-K63 and HA-Ub-K48 were co-expressed with TBK1 in HEK293T cells, the ubiquitination ladders of TBK1 were observed (Fig 5B, lanes 4 and 7). In the presence of BPLF1, both K63- and K48-linked ubiquitination of TBK1 was substantially diminished (Fig 5B, lanes 5 and 8). In contrast, both types of ubiquitination on TBK1 reappeared when C61A was expressed (Fig 5B, lanes 6 and 9). Plausibly, through its DUB activity, BPLF1 eliminates K63- and K48-linked ubiquitination on TBK1, resulting in the mitigation of TBK1 activation and signaling events.

## BPLF1 binds to STING, but not TBK1

In view of the apparent deubiquitination of STING and TBK1 by BPLF1, we further asked whether BPLF1 might interact with these effector proteins. A Myc-tagged version of BPLF1

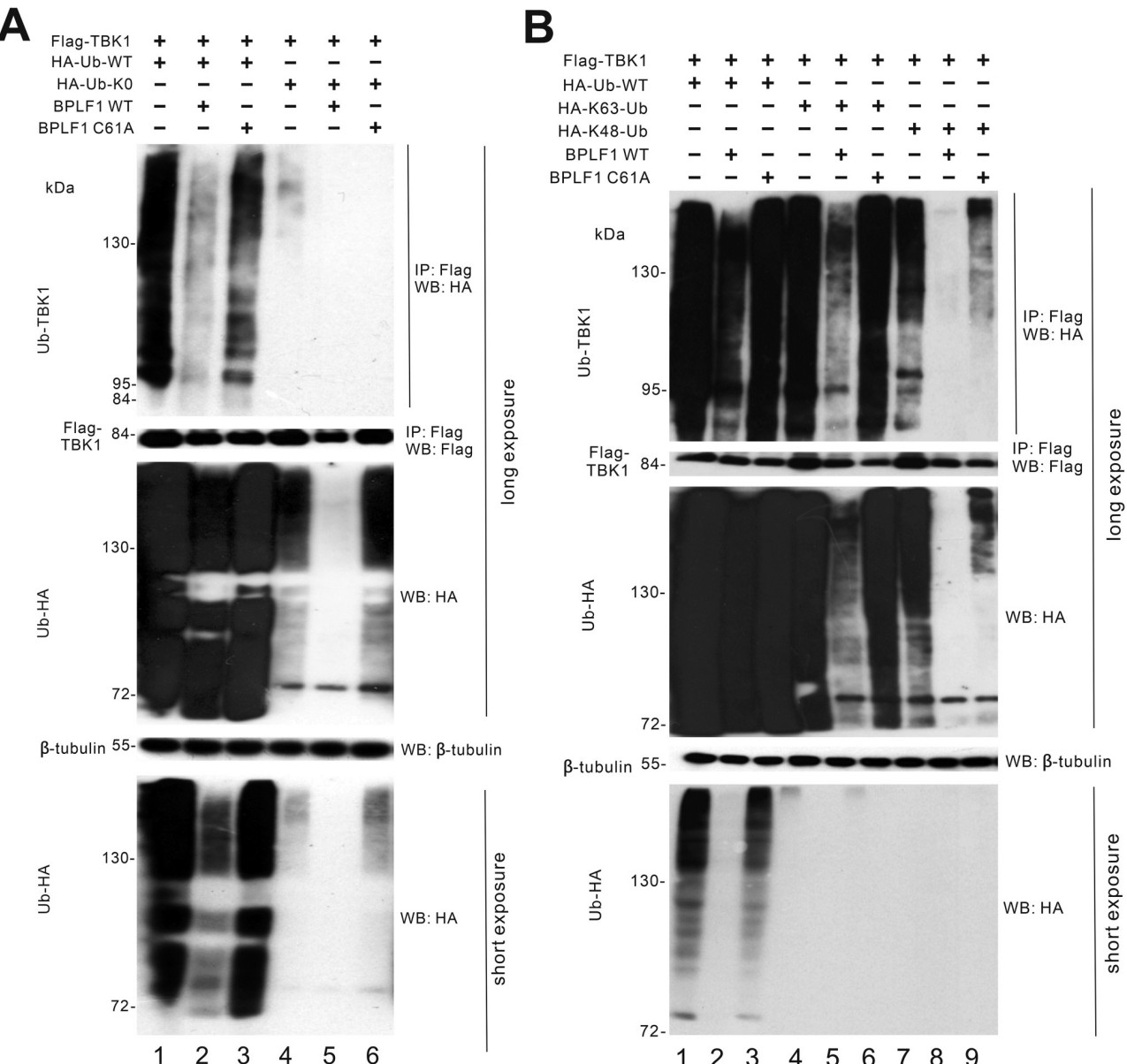

**Fig 5. BPLF1 perturbs TBK1 ubiquitination.** (A) Influence of BPLF1 and C61A on TBK1 ubiquitination. BPLF1, C61A and TBK1 expression plasmids were transfected into HEK293T cells together with HA-Ub-WT and lysine-free ubiquitin (HA-Ub-K0). (B) Influence of BPLF1 and C61A on TBK1 ubiquitination of different types. BPLF1, C61A and TBK1 expression plasmids were transfected into HEK293T cells together with HA-Ub-WT and HA-K63-Ub and HA-K48-Ub. Cells were harvested 24 hours after transfection for co-immunoprecipitation. The Flag-tagged TBK1 molecules were pulled down by anti-Flag antibodies. The bound fraction of the immunoprecipitates (IP) and the total lysates (input) were analyzed by Western blotting (WB) with anti-HA, anti- Flag and anti-β-tubulin antibodies. Long exposure was conducted for visualizing proteins in the upper panel and short exposure was conducted for visualizing proteins in the lower panel.

was employed here since both STING and BPLF1 were tagged with Flag and have similar protein sizes.

From the STING immunoprecipitates, the Myc-tagged BPLF1 can be detected (Fig 6A, lane 3). The immunoprecipitates were also probed with anti-Flag antibodies to verify the successful pull-down of Flag-STING protein (Fig 6A, lanes 2 and 3). In reciprocal immunoprecipitation experiment, when Myc-tagged BPLF1 was pulled-down, Flag-tagged STING can be detected

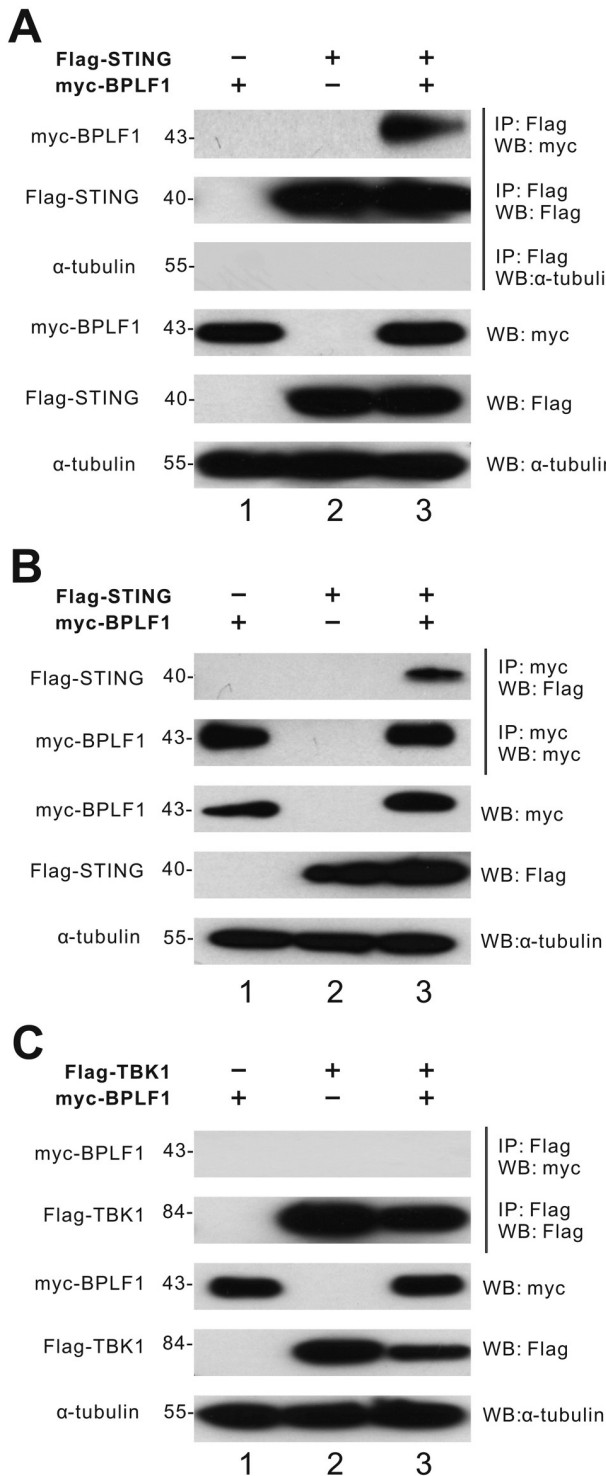

**Fig 6. BPLF1 binds to STING, but not TBK1.** (A, B) Co-immunoprecipitation of BPLF1 and STING. Myc-BPLF1 and Flag-STING expression plasmids were expressed in HEK293T cells. (C) No association between BPLF1 and TBK1. Myc-BPLF1 and Flag-TBK1 were expressed in HEK293T cells. Cells were harvested 24 hours after transfection for co-immunoprecipitation. The Flag-tagged STING and TBK1 proteins were pulled down by anti-Flag antibodies. The bound fraction of the immunoprecipitates (IP) and the total lysates (input) were analyzed by Western blotting (WB) with anti-Myc, anti-Flag and anti-α-tubulin antibodies.

in the BPLF1 immunoprecipitate (Fig 6B, lane 3). These results confirmed the association between BPLF1 and STING.

From the TBK1 immunoprecipitates, the anti-Myc antibody failed to detect any protein while Flag-tagged TBK1 was seen with anti-Flag antibody (Fig 6C, lane 3). Myc-tagged BPLF1 was instead found in the cell lysates. These results suggested that BPLF1 might not interact with TBK1, or their interaction would be too transient to be detected by immunoprecipitation.

## BPLF1 mitigates TBK1-induced dimerization of IRF3

To investigate the effect of BPLF1 on cGAS-STING-TBK1 pathway, the dimerization of IRF3 under TBK1 overexpression was analyzed by native PAGE. Crucial for IRF3 activation and signal transduction, dimerized IRF3 translocates into the nucleus to induce IFN-β gene transcription [54].

When Flag-tagged TBK1 and IRF3 were co-expressed in HEK293 cells, the dimerization of IRF3 can be observed in the native PAGE (Fig 7, lane 4). This TBK1-induced IRF3

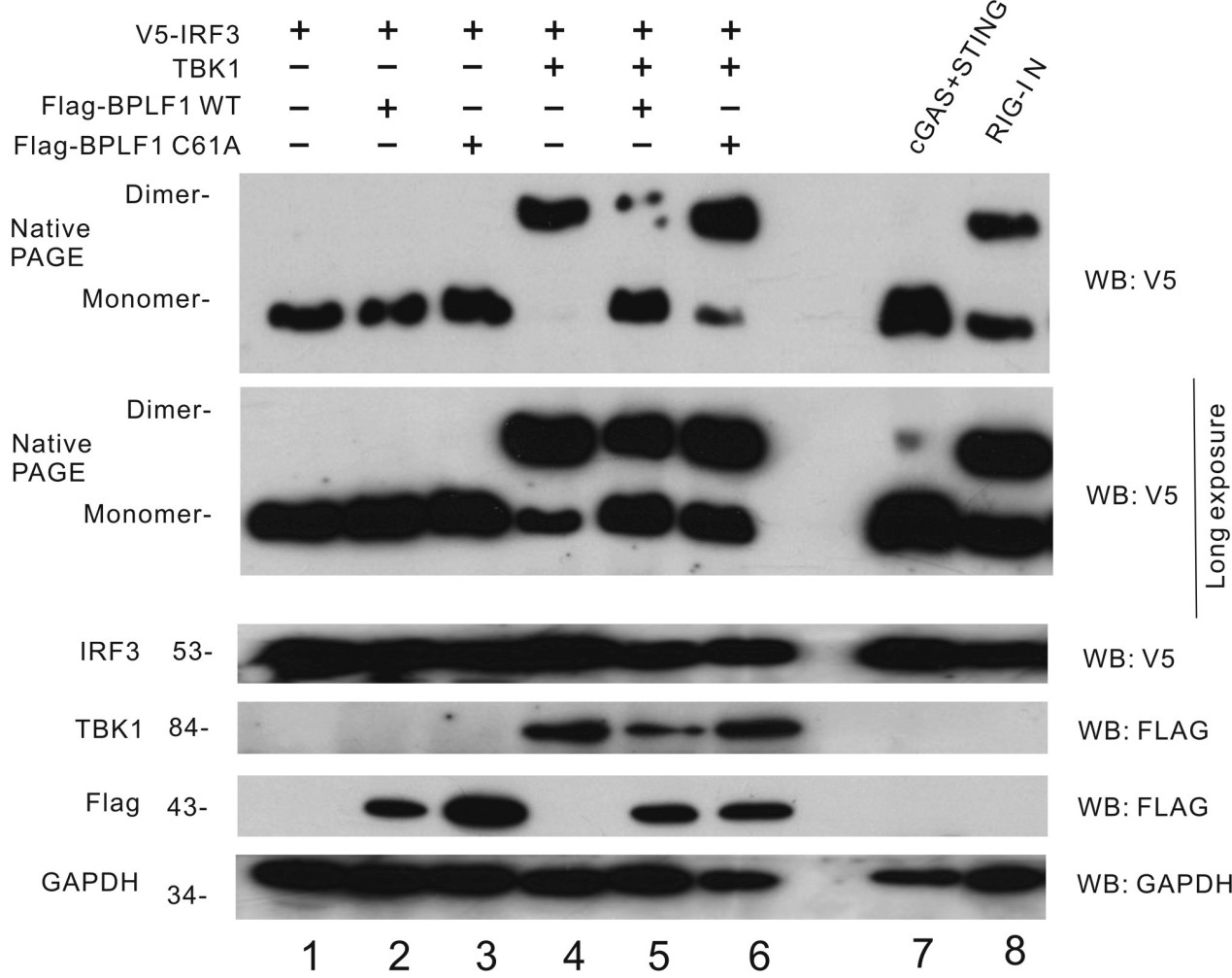

**Fig 7. BPLF1 mitigates TBK1-induced dimerization of IRF3.** IRF3 dimerization assay by native PAGE. BPLF1 and mutant C61A were transiently overexpressed in HEK293 cells together with TBK1 and V5-IRF3. Cells were harvested 48 hours after transfection. The cell lysates were subjected to SDS-PAGE and native PAGE. The lysates were analyzed by Western blotting by anti-V5, anti-Flag, anti-TBK1 and anti-α-tubulin antibodies.

dimerization significantly declined when BPLF1 was expressed in the cells (Fig 7, lane 5). When BPLF1 was replaced with C61A, no remarkable drop in IRF3 dimerization can be observed (Fig 7, lane 6). When we expressed cGAS + STING or RIG-I N in HEK293 cells, dimerization of IRF3 was observed, which served as the positive control and to demonstrate the induction of IRF3 dimerization upon activation of DNA sensing or RNA sensing (Fig 7, lanes 7 and 8). The total IRF3 protein expression levels were assessed by SDS-PAGE and no significant change was found (Fig 7). This result indicated that the DUB activity of BPLF1 would prohibit IRF3 dimerization, leading to mitigation of cGAS-STING-induced IRF3 activation.

## BPLF1 expression and IFN antagonism in different types of EBV⁺ cells

To derive additional insight on the IFN antagonism of BPLF1 during EBV infection of epithelial cells, we next set out to verify the expression patterns of BPLF1 in relation to other EBV proteins in different EBV-associated epithelial carcinoma cells. NPC43 are representative of EBV-associated nasopharyngeal carcinoma, while AGS-BX1 cells are gastric carcinoma cells carrying an EBV [55,56]. NP460EBV served as the reference to latent EBV infection in normal nasopharyngeal epithelial cells [57]. BPLF1 expression in HEK293p2089 and HEK293M81 cells [46,58,59] was also verified. All cells were treated to induce lytic EBV replication.

BPLF1 transcript was significantly induced when cells were treated with 12-O-tetradeca-noylphorbol-13-acetate (TPA) or TPA plus sodium butyrate for lytic induction (Fig 8A). LMP1 transcript was found in most EBV⁺ cells as control (Fig 8B). Interestingly, BPLF1 mRNA was also detected in AGS-BX1 cells and the latently infected NPC43 cells without treatment for lytic induction (Fig 8A). BPLF1 transcript was then normalized by LMP1 transcript to determine the relative expression level of BPLF1 and significantly higher relative BPLF1 transcript levels were still observed in AGS-BX1 cells without treatment for lytic induction (Fig 8C). Hence, BPLF1 might be expressed in gastric carcinoma cells, contributing to cancer development. The drug treatment induced lytic EBV replication in all types of cells, as indicated by the Zta and Rta transcript levels (Fig 8D and 8E). Interestingly, both Zta and Rta transcripts were detected in AGS-BX1 cells, and they were also found in small amounts in NPC43 cells. This was likely caused by spontaneous reactivation of EBV in these cells [60].

Considering that ample amount of BPLF1 transcript was detected in AGS-BX1 cells, it would be of interest to know whether BPLF1 expression contributes to the prohibition of IFN-β production in gastric carcinoma cells. To simulate the elevation of BPLF1 expression in gastric carcinoma, CRISPR-a was employed to induce BPLF1 expression in AGS-BX1 cells. When the dose of gRNA transfected into the AGS-BX1 cells was increased, the levels of IFN-β transcript induced by cGAS and STING overexpression were suppressed (Fig 8F). These results indicated that the expression of BPLF1 in EBV-infected gastric carcinoma cells would antagonize IFN-β production triggered by DNA sensing.

## BPLF1 suppresses IFN-β production in EBV-infected cells

To further study the biological effects of BPLF1 DUB during EBV infection, we constructed a recombinant EBV genome using bacterial artificial chromosome (BAC) recombineering. The EBV BAC p2089 [61] to be used for these experiments was carried in *E. coli* strain SW105 (SW105-p2089). After a two-step selection, SW105-p2089 clones were genotyped by PCR and Sanger sequencing (Fig 9A–9C). After genotyping, the mutant BAC genome was digested by BamHI restriction endonucleases to confirm the integrity of the EBV genome as previously described [62]. Similar digestion patterns from mutant clones were found when compared to

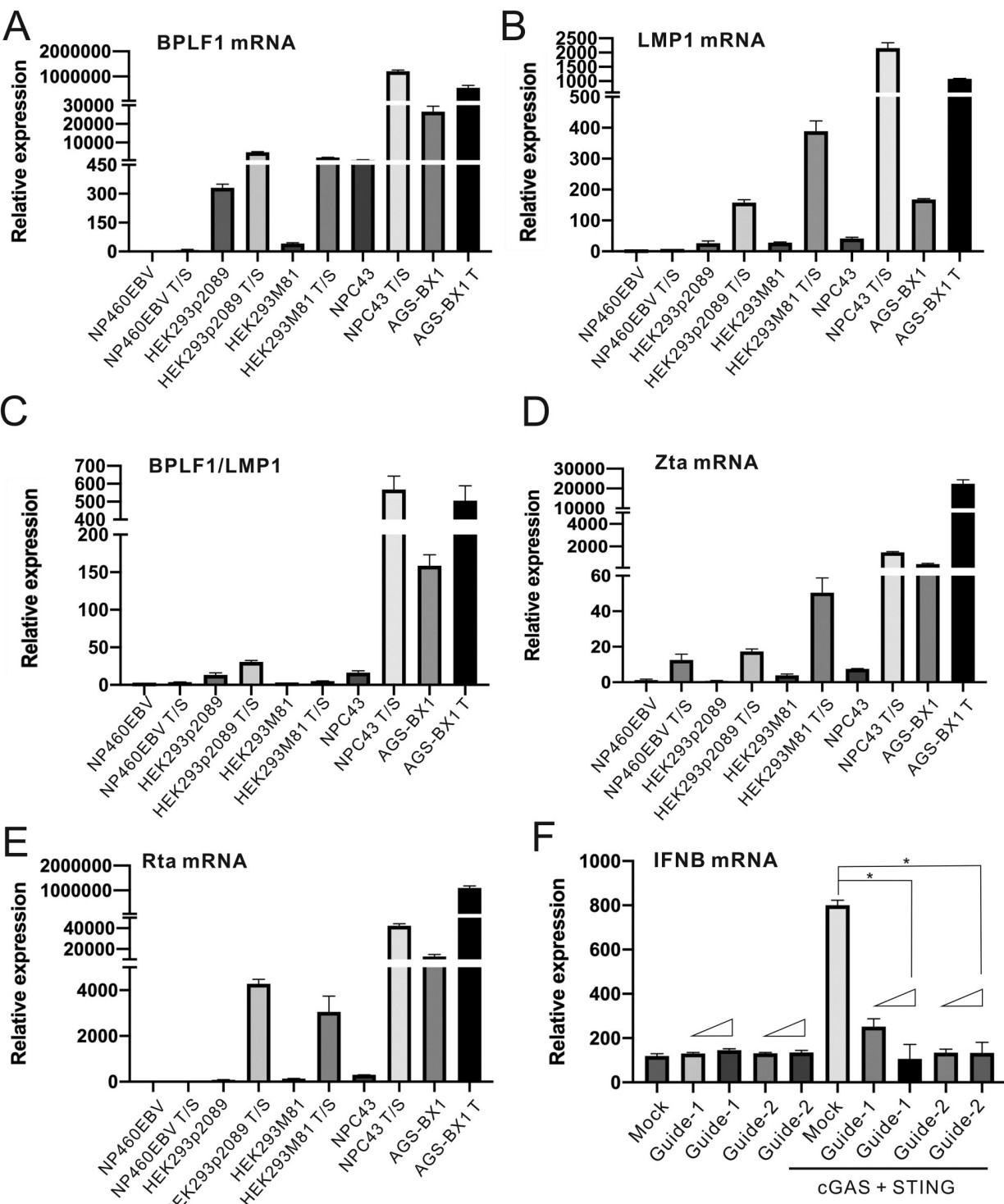

**Fig 8. Prevalence of BPLF1 in different EBV⁺ cells and the effect of BPLF1 activation on IFN-β level in gastric cancer cells.** (A-E) NP460EBV, HEK293p2089, HEK293M81, NPC43 and AGS-BX1 cells were seeded into 6-well plates. Cells were treated with either 40 ng/ml TPA (T) and 1.5 nM sodium butyrate (S) or 40 ng/ml TPA. The total cellular RNA was harvested 48 hours post-transfection and the BPLF1 (A), LMP1 (B), Zta (D) and Rta (E) transcript expression levels were determined by RT-qPCR. The relative expression levels were normalized to endogenous GAPDH transcript expression levels. (C) The relative expression levels of BPLF1 transcript were further normalized with the relative expression levels of LMP1 transcript. (F) AGS-BX1 cells were transfected with cGAS and STING expression plasmids, lentiSAMv2 carrying guide-1 and guide-2, and lentiMPHv2. Total cellular RNA was harvested 48 hours post-transfection and the IFN-β transcript expression levels were measured by RT-qPCR.

The IFN-β transcript expression levels were normalized to the endogenous GAPDH transcript level. The mean values of three biological replicates (n = 3) were represented by the bars and their respective standard deviations were depicted as the error bars. The statistical significance among selected samples were analyzed using two-tailed Student's t-test for paired samples and the range of $p$ values was indicated (*: $p < 0.05$).

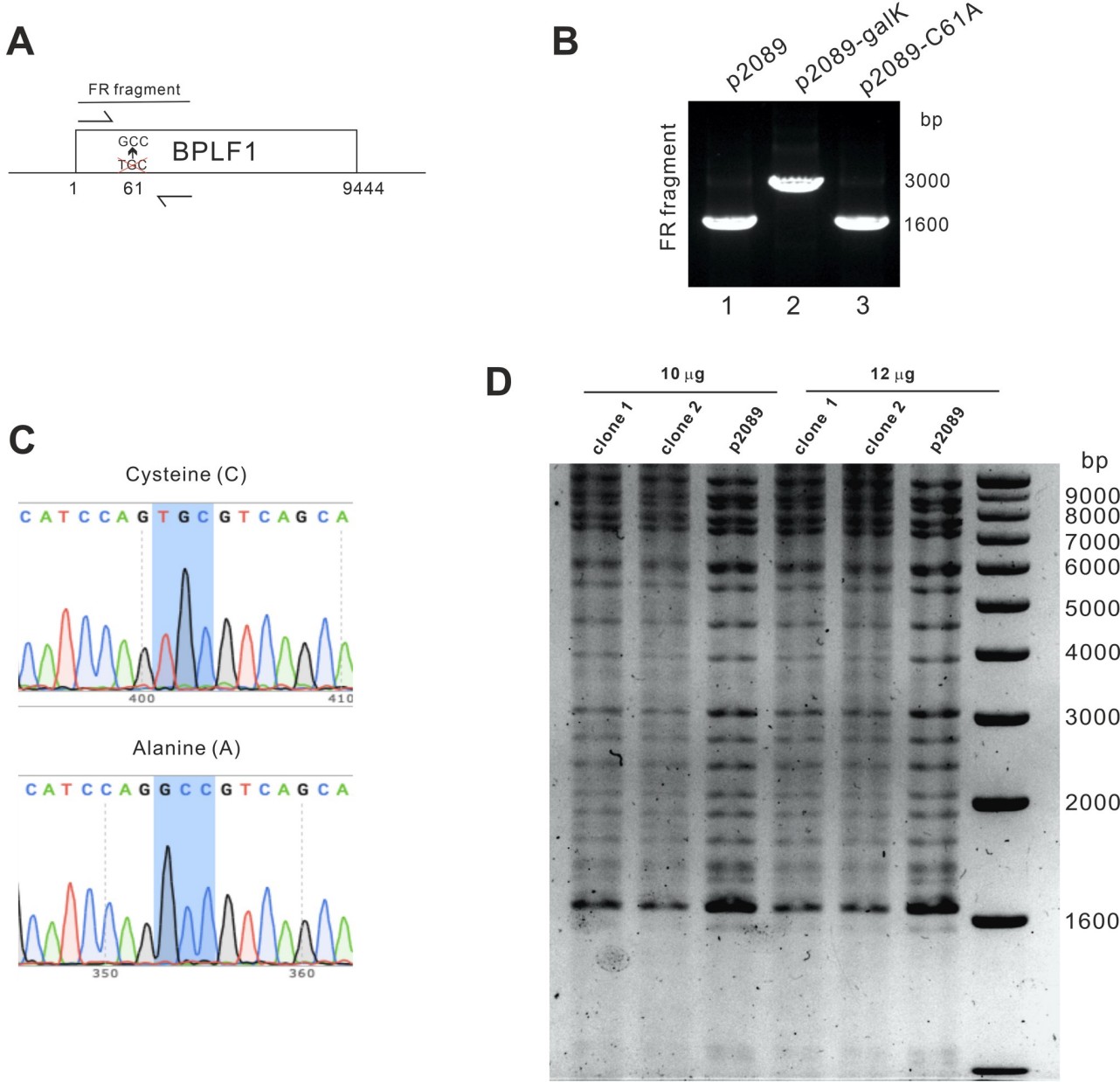

**Fig 9. Characterization of recombinant EBV with a catalytically dead BPLF1.** (A) Schematic diagram of the flanking primers designed for PCR genotyping of the fragments around position 61 at the N-terminus of BPLF1 in EBV BAC. (B) PCR amplification of the fragments of unedited SW105-p2089 clone, *galK* inserted clone and C61A mutant clone. (C) Sequencing results of the unedited SW105-p2089 clone and C61A mutant virus clone. (D) BamHI restriction patterning of mutant C61A cloned 1 and 2 as well as unedited EBV BAC SW105-p2089. 10 μg and 12 μg of the EBV BACs were digested and the BamHI restriction patterning was evaluated. Nanopore sequencing of the BACs has been conducted and no undesirable change was spotted in the mutant virus.

that of SW105-p2089 WT (Fig 9D). Thus, EBV BAC clones with a DUB-dead mutation in BPLF1, designated SW105-p2089-C61A, were successfully generated.

SW105-p2089-C61A clones were transfected into HEK293 cells. Stable cell lines were selected with 100 μg/ml hygromycin B for a period of 21 days [63]. The GFP$^+$ single clones were then picked and purified. Ample amounts of BPLF1 transcript were detected in all stable cells (Fig 10A). BPLF1 transcript was also found in two clones of HEK293p2089 cells (Fig 10A). Thus, BPLF1 was abundantly expressed in the HEK293p2089 and HEK293p2089-C61A cells. Other lytic EBV gene products including Zta, Rta and BMRF2 were also measured for both HEK293p2089 and HEK293p2089-C61A cells. Both wild type and mutant cell clones expressed a small amount of these lytic gene products. The levels of Zta and BMRF2 transcripts were higher in HEK293p2089-C61A cells. Nevertheless, all three lytic gene products showed a drastic increase upon treatment with TPA and sodium butyrate (S3A–S3C Fig). These results indicated the functionality of the HEK293p2089-C61A cells with the mutation.

The ubiquitination patterns of STING in the HEK293p2089 and HEK293p2089-C61A cells were then assessed by co-immunoprecipitation. cGAS, Flag-tagged STING and HA-Ub-WT were co-expressed in these cells. In the STING immunoprecipitate from cGAS-STING-overexpressing HEK293 cells, the ubiquitin ladder was evident. The ubiquitin ladder from the STING immunoprecipitate was diminished in cGAS-STING-overexpressing HEK293p2089 cells (Fig 10B, lane 2 vs 1). In contrast, from the immunoprecipitates of STING from HEK293p2089-C61A cells, no decline in ubiquitination level was observed (Fig 10B, lanes 3–6 vs. 1). These results suggested deubiquitination of STING by BPLF1 in the EBV-infected cells and reversion of this phenotype when BPLF1 DUB domain was inactivated.

The impact of the BPLF1 DUB domain on IFN-β production in these EBV-infected cells were further assessed. HEK293, HEK293p2089 and HEK293p2089-C61A cells were first assessed using the IFNB-Luc reporter. Upon overexpression of cGAS and STING, the IFN-B-Luc activity was induced in HEK293 cells (Fig 10C). However, this induction was much less pronounced in the two clones of HEK293p2089 cells (Fig 10C). When cGAS and STING were expressed in the four clones of HEK293p2089-C61A cells, no significant drop in IFNB-Luc activity was observed (Fig 10C). A similar pattern was also when the IFN-β transcript was measured (Fig 10D). These results showed that BPLF1 DUB antagonizes IFN-β productions during EBV infection.

Representative clones HEK293p2089 and HEK293p2089-C61A were subjected to further analysis of IFNB-Luc activation. The basal IFNB-Luc activity was higher in HEK293p2089-C61A cells than in HEK293p2089 cells (Fig 10E), compatible with the IFN antagonism of BPLF1. When the clones were induced with TPA to undergo lytic replication, IFNB-Luc activity was blunted (Fig 10E). In the presence of TBK1 or cGAS + STING, IFN-B-Luc activity was much induced particularly in the HEK293p2089 cells. However, lytic induction with TPA suppressed IFN-β production in the same settings, but the suppression was incomplete in most cases (Fig 10E). Plausibly, BPLF1 cooperates with other EBV-encoded IFN antagonists to counteract TBK1- and cGAS-SING-induced activation of IFN-β production. Taken together, in EBV-infected cells, BPLF1 suppressed IFN-β production induced by DNA sensing through deubiquitination of the adaptor protein STING.

To confirm the biological importance of the deubiquitinating ability of BPLF1 on EBV infectivity, the WT and mutant p2089 viruses were produced from the cells. Both viruses were then used to infect HEK293 cells expressing either cGAS + STING or TBK1 to interrogate the requirement for the DUB domain of BPLF1. When cells expressed either cGAS + STING or TBK1, a more pronounced drop in the percentage of GFP$^+$ cells was observed (S4A Fig). When BPLF1 was also expressed in the cells, a higher percentage of GFP$^+$ cells was seen (S4A Fig). The IFN-β mRNA levels of the infected cells were also measured. For cells infected with

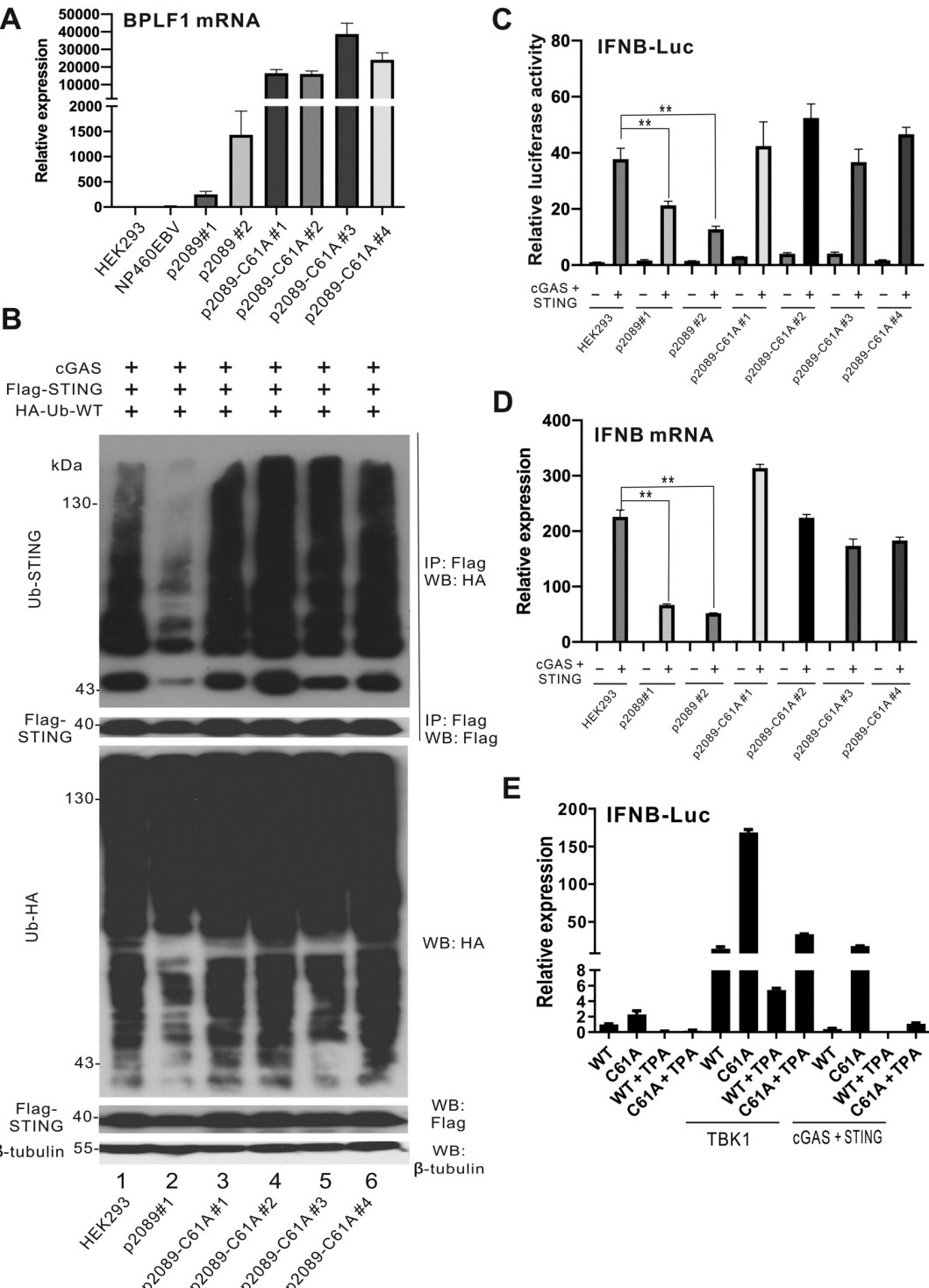

**Fig 10. BPLF1 suppresses IFN-β production in EBV-infected cells.** (A) BPLF1 mRNA expression in EBV$^+$ cells. HEK293, NP460EBV, HEK293p2089#1–2, HEK293p2089-C61A#1–4 cells were seeded into 6-well plates. The total cellular RNA was harvest 48 hours after transfection and the BPLF1 transcript expression levels were determined by RT-qPCR and the relative BPLF1 transcript expression levels were normalized to endogenous GAPDH transcript levels. (B) STING ubiquitination in cells carrying BPLF1-inactive EBV. HEK293, HEK293p2089 and HEK293p2089-C61A#1–4 cells were seeded into 60mm dishes. Cells were harvested 48 hours after transfection for co-immunoprecipitation. The Flag-tagged STING molecules were pulled down by the anti-Flag antibodies. The bound fraction of the immunoprecipitates (IP) and the total lysates (input) were analyzed by Western blotting (WB) with anti-HA, anti-Flag and anti-β-tubulin antibodies. (C) IFNB-Luc activation in cells

carrying BPLF1-inactive EBV. IFNB-Luc and TK-Luc reporter plasmids were transfected into HEK293, HEK293p2089#1–2, HEK293p2089-C61A#1–4 cells together with cGAS and STING expression plasmids. (D) IFN-β induction in cells carrying BPLF1-inactive EBV. Cells were harvested 24 hours post-transfection for dual-luciferase assay. HEK293, HEK293p2089#1–2, HEK293p2089-C61A#1–4 cells were transfected with cGAS and STING expression plasmids. The total cellular RNA was harvested 48 hours post-transfection and the IFNβ transcripts expression levels were measured by RT-qPCR. The IFN-β transcript expression levels were normalized by the endogenous GAPDH transcripts level. The mean values of three biological replicates (n = 3) were represented by the bars and their respective standard deviations were depicted as the error bars. The statistical significance among selected samples were analyzed using two-tailed Student's t-test for paired samples and the p-values were indicated. (E) IFNB-Luc activation in cells stably carrying BPLF1-inactive EBV. WT p2089 and BPLF1C61A p2089 EBV BAC were transfected into HEK293 cells for stable cell construction. The transfected cells were selected with hygromycin for about a week and the survival cells were then subject for IFN-β analysis. The stable cells were transfected with TBK1 or cGAS and STING for IFN-β production. TPA is added at 24 hours post-transfection to stimulate the lytic transcript production. Cells were harvested in the next day and RT-qPCR were performed to measure the IFN-β transcript level. GAPDH was used for normalization.

p2089-C61A virus, a higher level of IFN-β mRNA was detected (S4B Fig). These results indicated the requirement of the deubiquitinating ability of BPLF1 for not only viral infectivity but also viral suppression of cGAS-STING- and TBK1-induced IFN-β production during primary infection.

## Discussion

In our present study, EBV large tegument protein BPLF1 was identified as a virulence factor that mitigates cGAS- and RIG-I-induced IFN-β production (Figs 1 and 2). BPLF1 acts upstream of IRF3 and in a DUB-dependent manner (Figs 2 and 3). BPLF1 DUB removes all types of ubiquitin moieties on STING and TBK1 (Figs 4–6) to prevent IRF3 dimerization (Fig 7). Induction of BPLF1 expression from EBV genome by CRISPR-a suppressed cGAS-STING-induced IFN-β expression (Figs 2 and 8). Finally, DNA-induced STING ubiquitination and IFN-β gene transcription were more prominent in HEK293 cells stably harboring a recombinant EBV with a DUB-dead mutation in BPLF1 (Figs 9 and 10). The deubiquitinating domain of the virus was crucial for viral infectivity and viral suppression of IFN-β production (S4 Fig). These findings provide a new mechanism by which EBV BPLF1 antagonizes host innate immunity.

EBV BPLF1 is a conserved DUB in herpesviruses and its orthologs include ORF64 in Kaposi sarcoma-associated herpesvirus (KSHV), UL48 in HSV-1 and UL48 in cytomegalovirus. All these herpesviral DUBs contain a highly conserved catalytic triad [32]. Indeed, ORF64 from MHV68 and its homologs from KSHV, HSV-1 and mouse cytomegalovirus were found to deubiquitinate STING to prevent STING activation [63]. It will be of interest to see whether other herpesviruses might also employ their DUBs to circumvent cGAS- and RIG-I-dependent IFN production.

In our study, the suppressive effects of BPLF1 on IFN production were shown by overproduction of BPLF1 325 and induced expression of full-length BPLF1 from EBV genome by CRISPR-a (Figs 2 and 8). BPLF1 325 has been widely adopted in the study of BPLF1 DUB [34,38,64]. Expression of full-length BPLF1 was verified by RT-qPCR due to the lack of anti-BPLF1 antibodies. In our experiments, BPLF1 325 demonstrated more robust activity than full-length BPLF1, which may be explained by the existence of potential inhibitory domain in full-length BPLF1 and/or limited expression level of full-length BPLF1 induced by CRISPR-a. Nevertheless, the IFN antagonism of BPLF1 was consistently supported by multiple lines of evidence including luciferase reporter assays, RT-qPCR, ubiquitination assays and BAC recombineering.

Results from our co-immunoprecipitation assays (Figs 4 and 5) were consistent with the notion that BPLF1 removes all types of ubiquitin moieties on STING and TBK1.

Ubiquitination of STING and TBK1 was remarkably diminished in the presence of BPLF1, but partially restored upon expression of the DUB-dead mutant C61A (Fig 4). The effector proteins in the innate immune pathways are tightly regulated by various PTMs and ubiquitination plays a major role in such regulation [17]. Whereas K63- and K27-linked ubiquitination of STING is important for its activation, K48-linked ubiquitination triggers its proteasomal degradation [18,20,65]. Our results indicated that STING turnover was unaffected in the presence of BPLF1 (Fig 4). Thus, the net result of perturbing all three types of polyubiquitination by BPLF1 would be the attenuation of STING activity during viral infection. For TBK1, there are two main types of ubiquitination that is K63- and K48-linked. Both types of were found to be disturbed in the presence of BPLF1 (Fig 5). Whereas the K63-linked ubiquitination of TBK1 is required for TBK1 activation, the K48-linked ubiquitination triggers TBK1 proteasomal degradation. TBK1 turnover was minimally affected when BPLF1 was expressed (Fig 5). The diminution of both types of ubiquitination on TBK1 would render the TBK1 inactive. Since TBK1 is a common adaptor protein for both DNA sensing and RNA sensing pathways, the mitigation of TBK1 ubiquitination by BPLF1 would antagonize both DNA sensing and RNA sensing [66,67]. Reasonably, the prevention of IRF3 dimerization by BPLF1 (Fig 7) reflects its suppressive effects on STING and TBK1.

The results from luciferase assay and co-immunoprecipitation experiments for cells stimulated with ISD90 demonstrated the effect of BPLF1 on cGAS-STING signaling pathway induced by dsDNA ligands (S2A and S2B Fig). BPLF1 was shown to mitigate the ISD90-induce IRF3-Luc activities and abolish the ISD90-induced ubiquitinations on endogenous STING protein. Replacing BPLF1 with BPLF1-C61A allow partial restoration in IRF3-Luc activities and the ubiquitination on endogenous STING. These assays demonstrated the potent suppression by the DUB activities of BPLF1 on the ligand-stimulated cGAS-STING signaling pathway.

We showed by reciprocal co-immunoprecipitation and immunoblotting the binding of BPLF1 to STING but not TBK1 (Fig 6). Plausibly, BPLF1 binds to STING to exert its DUB activity. In contrast, the lack of interaction between BPLF1 and TBK1 raises several possibilities. First, BPLF1 might weakly or transiently interact with TBK1. Second, BPLF1 could perturb TBK1 ubiquitination through an indirect route, such as via a third partner. Further investigations are required to fully elucidate how BPLF1 deubiquitinates TBK1.

We characterized the IFN antagonism of BPLF1 by creating a recombinant EBV with a DUB-dead BPLF1. We noted the high abundance of BPLF1, Zta and BMRF2 transcripts in HEK293p2089-C61A stable cells (Fig 10 and S3 Fig). Clonal effect was minimized by using 2–4 clones of HEK293p2089 and HEK293p2089-C61A cells. It would be of interest to determine whether the underexpression of BPLF1 and other lytic transcripts in HEK293p2089 cells might be attributed to epigenetic silencing of the EBV genome after long-term passage [68]. The prominence of STING ubiquitination and IFN-β production in HEK293p2089-C61A cells compared to HEK293p2089 cells (Fig 10) lent crucial support to the IFN antagonism of BPLF1 in the context of EBV infection. STING ubiquitination and IFN-β production were compromised even when BPLF1 expression was low in HEK293p2089 cells, and they were fully restored in HEK293p2089-C61A cells with high expression of C61A. Together with results from the infection experiments with enforced overexpression of BPLF1 (Fig 3), these data demonstrated the biological importance of BPLF1 DUB during EBV infection. Furthermore, the results from the infection experiments for wild type p2089 and mutant p2089C61A viruses has also demonstrated the biological significance of BPLF1 DUB during EBV infection (S4 Fig). BPLF1 has also been shown to promote EBV replication and evade innate immunity through other mechanisms [35–43]. The IFN antagonism of BPLF1 demonstrated in our study is generally consistent with its role in viral replication and immune evasion.

EBV has evolved multiple approaches in innate immune evasion. EBV-encoded BLRF2, a homolog of KSHV's ORF52, was proposed to bind to cGAS like ORF52 so as to prohibit cGAS activation [69]. EBV BGLF4 was found to diminish the ability of IRF3 to induce IFN-β gene transcription [70]. Here we reported that EBV would also interfere with STING and TBK1 to halt the production of IFN-β. We also performed a comprehensive functional screening for IFN-β antagonists upon stimulation with cGAS and STING (Fig 1). Stimulation of IFNB-Luc activity was still seen upon lytic induction of HEK293p2089-C61A cells (Fig 10), indicating the existence of non-BPLF1 IFN antagonists that counteract TBK1 or cGAS-STING signaling. Plausibly, BPLF1 cooperates with additional EBV-encoded IFN antagonists to achieve optimal suppression of TBK1 and cGAS-STING pathways.

Lytic gene expression was more robust in HEK293p2089-C61A cells if compared to HEK293p2089 cells (Fig 10). Plausibly, the HEK293p2089-C61A cells were newly established and the mutant virus might need more time to adapt to the cellular environment. The viral genome might not as hypermethylated as that of WT p2089, leading to the different pattern of lytic gene expression.

cGAS-STING-dependent DNA sensing is a double-edged sword for the host to fight against cancer. The activation or suppression of this pathway might contribute to the development of various cancers [71]. For EBV oncogenesis, only LMP1 has been well established as an onco-protein that contributes to the development of B cell lymphoma. The mechanism by which EBV drives carcinogenesis in nasopharyngeal and gastric carcinomas has remained largely elusive [72]. Although EBV infection exhibits a certain type of latency program in a specific type of cells, the latency program may change from time to time, which also contributes to oncogenesis or metastasis [73]. Lytic replication is not only required for EBV virion production, but also plays a critical role in EBV carcinogenesis [4,74].

Ubiquitination has been reported to be an important PTM for tumor suppressor genes in nasopharyngeal and gastric carcinomas [75]. For EBV+ nasopharyngeal carcinoma cells, somatic mutations have been found in the cellular DUB gene known as CYLD. CYLD is a tumor suppressor gene and a negative regulator of NF-κB pathway. The mutations on these genes contribute to better tumor growth and metastasis [75–77]. Meanwhile, we detected BPLF1 transcript in NPC43 cells (Fig 8), which might be caused by spontaneous lytic reactivation of the virus in a subset of nasopharyngeal carcinoma cells. BPLF1 has been shown to antagonize NF-κB activation by deubiquitinating multiple effectors in TLR signaling pathway [31]. The presence of BPLF1 in nasopharyngeal carcinoma cells suggested that the virus might also employ deubiquitination to modify the cellular environment for optimal replication. Deletion of cellular DUB genes might counteract BPLF1-mediated deubiquitination to maintain normal cellular function. It would be of great interest to delineate whether compromising the cellular DUBs might compensate for the potent DUB activity of BPLF1 in EBV+ nasopharyngeal carcinoma cells.

For EBV+ gastric carcinoma, EBV might also participate in tumorigenesis by altering the cGAS-STING pathway. MUS81 overexpression has been proposed to contribute to the tumorigenesis and metastasis of gastric carcinoma [78]. Anti-MUS81 therapy has been developed to induce the accumulation of cytosolic DNA so as to activate the cGAS-STING pathway for anticancer effects. Abundant expression of BPLF1 in AGS-BX1 cells (Fig 8) suggested efficient suppression of the cGAS-STING pathway in gastric carcinoma by EBV. With reference to the promising results from the anti-MUS81 therapy on gastric carcinoma, it would be interesting to know whether combining anti-MUS81 with anti-BPLF1 treatment would produce optimal anticancer effects in gastric carcinoma.

Furthermore, BPLF1 might also affect tumorigenesis by suppressing tumor suppressor DAPK3. DAPK3, also known as ZIPK, is a tumor suppressor that activates antitumor

immunity via STING. DAPK3 would induce K63-linked ubiquitination and inhibit K48-linked ubiquitination on STING to induce STING-mediated anti-tumor responses [79]. DAPK3 was found to be frequently mutated in gastric carcinoma [80]. In EBV$^+$ gastric carcinoma, the anti-tumor effects of DAPK3 may be even more limited given that BPLF1 would counteract DAPK3-mediated STING activation. Hence, BPLF1 may contribute to gastric carcinogenesis and metastasis by acting as DAPK3 agonist. Further investigations are required to clarify the role of BPLF1 and its interaction with other cellular tumor suppressor genes in the context of EBV carcinogenesis.

## Materials and methods

### Plasmids

The expression plasmids of cGAS, STING, RIG-I N, TBK1, IRF3, IRF3-5D, IκB-sr, HA-Ub-WT and HA-Ub-K0, and luciferase reporter plasmids IFNB-Luc, IRF3-Luc, TK-Luc have been described previously [44,46,81]. The EBV protein expression library was kindly provided by Lori Frappier from University of Toronto, and the p2089 bacmid was provided by Wolfgang Hammerschmidt from German Center for Infection Research (Munich, Germany). The expression constructs of BPLF1 mutant C61A, HA-Ub-K63, HA-Ub-K48, HA-Ub-K27, HA-Ub-K63R, HA-Ub-K48R were constructed by Q5 Site-Directed Mutagenesis Kit (New England Biolabs). Lenti-MPHv2 and lenti-SAMv2 plasmids [45] were gifts from Feng Zhang from Broad Institute.

### Antibodies

Mouse anti-Myc (9E10), rabbit anti-HA (F-7), mouse anti-α-tubulin (B-7) and mouse anti-β-tubulin (3F3-G2) antibodies were purchased from Santa Cruz. Mouse anti-Flag (F3165), rabbit anti-Flag (F7425) and mouse anti-ubiquitin (P4D1) antibodies were purchased from Sigma-Aldrich. Rabbit anti-cGAS (D1D3G), rabbit anti-STING (D2P2F) and rabbit anti-TBK1 (D1B4) antibodies were from Cell Signaling Technology. Mouse anti-V5 (E10/V4RR) antibodies were from Invitrogen.

### Digestion, ligation and transformation

The dsDNA insert oligos and the vector plasmids were digested with restriction enzymes at 37˚C overnight. The digested inserts and vectors were mixed in a 3:1 molar ratio and incubated with T4 DNA ligase (Life Technologies) for 15 min at room temperature for ligation. The ligated constructs were then mixed with 100 μl competent DH5α strain of *Escherichia coli* and rested on ice for 30 min. After that, the *E. coli* was heat-shocked at 42˚C for 90 sec and chilled on ice for 3 min. The transformed *E. coli* was recovered in 1 ml Luria Bertani (LB) broth at 37˚C for 45 min before spreading onto LB-agar plate.

### Plasmid preparation

The transformed bacteria were plated on a LB-agar (Sigma plates) with 100 μg/ml ampicillin for overnight incubation at 37˚C. Positive clones were picked and cultured in LB Broth (with ampicillin) for 4 hours. The clone will be sequenced by Sanger sequencing. After the DNA sequence was verified, the bacterial culture was expanded to 100 ml LB Broth (with ampicillin). The DNA plasmids were harvested with QIAGEN Plasmid Midi Kit (QIAGEN).

## BAC recombineering

Procedures of BAC recombineering have been described previously [46,81]. Here, two galK primers with 50bp homology arm flanking the desired mutation site were designed (forward primer: 5′-CCTCGTGCAA CCAGGCCCAC TGCAAGTTTG GCCGCTTTGC CGGCATC CAG CCTGTTGACA ATTAATCATC GGCA-3′, reverse primer: 5′-CGGCCGGCCA GGAAGCTTTT GACCAGGTAG AGGACGCAGT TGCTGACTCA GCACTGTCCT GCTCCTT-3′). Another pair of oligos were designed with 100 bp complementary oligos with point mutation at position 61 of BPLF1 (C61A forward: 5′ –CCTCGTGCAA CCAGGCCCAC TGCAAGTTTG GCCGCTTTGC CGGCATCCAG GCCGTCAGCA ACTGCGTCCT CTACCTGGTC AAAAGCTTCC TGGCCGGCCG-3′; C61A reverse: 5′–CGGCCGGCCA GGAAGCTTTT GACCAGGTAG AGGACGCAGT TGCTGACGGC CTGGATGCCG GCAAAGCGG CCAAACTTGC AGTGGGCCTG GTTGCACGAG G-3′).

## Cell culture

Human embryonic kidney cell line (HEK293), HEK293T, HEK293 stably harboring EBV M81 BAC (HEK293M81), HEK293 stably harboring EBV p2089 BAC (HEK293p2089) and AGS-BX1 cells were cultured in RPMI 1640 medium (Gibco, Life Technologies) supplemented with 10% fetal bovine serum (FBS; Gibco, Life Technologies) and 10% penicillin-streptomycin (Pen-Strep; Thermo Fisher) at 37˚C in a 5% $CO_2$ atmosphere. NPC43 cells were cultured in RPMI 1640 supplemented with 10% FBS, 10% Pen-Strep and 0.2% Rho kinase inhibitor Y-27632 (ROCK inhibitor; Tocris Bioscience). The stable HEK293p2089-C61A cells were cultured in RPMI 1640 supplemented with 10% FBS, 10% Pen-Strep and 100 µg/ml hygromycin B (Thermo Fisher).

## Transient transfection

HEK293, HEK293T, HEK293M81, HEK293p2089 and HEK293p2089-C61A cells were seeded onto 60mm dishes, 6-well or 24-well plates (Iwaki) at a concentration of $1 \times 10^5$ per ml and transfected with GeneJuice (Novagen) 24 hours later in a ratio of 1µg DNA to 3µl GeneJuice. ISD90 (4 µg/ml) was transfected into HeLa cells in a 1 µg DNA to 3 µl Lipofectamine 3000/ P3000 ratio according to manufacturer's instruction. ISD90 primers were 5′- TACAGATCTA CTAGTGATCT ATGACTGATC TGTACATGAT CTACATACAG ATCTACTAGT GATC TATGAC TGATCTGTAC ATGATCTACA (forward) and 5′- TGTAGATCAT GTACAGA TCA GTCATAGATC ACTAGTAGAT CTGTATGTAG ATCATGTACA GATCAGTCAT AGATCACTAG TAGATCTGTA (reverse).

## Dual luciferase reporter assay

HEK293 cells were seeded on 24-well plates (Iwaki) and transfected with designated plasmids. After 24 hours post-transfection, cells were lysed in 1× Passive Lysis Buffer (PLB; Promega). Each well was added with 100µl 1×PLB and lysed for 30 min at room temperature. 30µl of cell lysate was then transferred to the 96-well plate. The *Firefly* and *Renilla* luciferase activities were measured by the Microplate Luminometer LB 96V (EG&G Berthold, MicroLumat Plus) by Dual-luciferase Reporter Assay System (Promega).

## Co-immunoprecipitation

Cells were lysed using RIPA buffer (50mM Tris-Cl, pH7.4, 150mM, NaCl, 0.5mM EDTA, 1% NP40, 1% Triton X-100, 1% sodium deoxycholate, 0.1% SDS and 10mM N-ethylmaleimide) added with Biotool Protease Inhibitor Cocktail (biotool.com). 400µl of RIPA was added to the

60mm cell culture dish for 30 min incubation at 4˚C. The lysate was centrifuged at 13500 rpm for 15 min at 4˚C and the supernatant was transferred to a new Eppendorf. The protein concentration of the supernatant was measured using Bradford solution. 15μl of Dynabeads M-280 sheep anti-rabbit IgG (Thermo Fisher Scientific) was rinsed with 500μl PBS and then incubated in 500μl 1% bovine serum albumin (BSA; Sigma) at 4˚C to roll for 10 min. 1.5μg antibody was then applied to the beads for 1 hour rolling at 4˚C. The beads were then washed with 500μl 1%BSA. Lysate with 1mg proteins will be topped up to 500μl in total volume with RIPA and the lysate would be applied to the beads to roll for 1–2 hour(s). After incubation, the beads were rinsed with 500μl RIPA for three times. 30μl 2× Protein Sample Buffer (0.1M Tris-Cl, pH 6.8, 2.5% SDS, 20% glycerol, 25% β- mercaptoethanol, and 0.05% bromophenol blue) was added to re-suspend the beads. The immunoprecipitation samples were then boiled for 10 min.

## Protein sample preparation for SDS-PAGE

Cells were lysed in RIPA buffer supplemented with protease inhibitor. 400μl of RIPA was added to each 60 mm dish for 30 min incubation at 4˚C. The cell lysates were centrifuged at 13500 rpm for 15 min at 4˚C and the supernatants were transferred to a new Eppendorf. The protein concentration is measured using Bradford assay. 6× Protein sample buffer was added to the lysate and the protein sample was boiled for 10 min.

## Protein sample preparation for native PAGE

HEK293 cells were lysed in native gel lysis buffer (50mM Tris-Cl at pH7.4, 150mM NaCl and 1% NP-40) supplemented with protease inhibitor. 200μl native gel lysis buffer was added to each well in 6-well plates (Iwaki) and incubated at 4˚C for 30 min. The cell lysates were then centrifuged at 13500 rpm for 15 min at 4˚C. The supernatants were transferred to a new Eppendorf with its protein concentration measured by Bradford assay. 2× native gel sample buffer was added to the lysate before applying the lysates to native PAGE analysis.

## SDS-PAGE and electroblotting

The protein samples and immunoprecipitation samples were boiled for 10 min before applying to SDS-gel. The denatured samples were applied to 8–15% discontinuous polyacrylamide gel (stacking gel at pH6.8; resolving gel at pH8.8) in 1 × SDS-PAGE running buffer (25mM Tris, 192mM glycine, 0.1%SDS, pH8.3). A constant current of 25mA per gel was applied for electrophoresis. After electrophoresis, the protein in the polyacrylamide gel was transferred to a Immobilon-P polyvinylidene difluoride (PVDF) membrane (Millipore) through electroblotting. The electroblotting was conducted in a Semi-dry Hoefer miniVE apparatus (Hoefer SemiPhor, Amersham). Electroblotting was conducted in semi-dry transfer buffer (25mM Tris, 192mM glycine, 20% methanol, pH8.3) at 150mA per gel for 70 min.

## Native PAGE and electroblotting

A 10% discontinuous polyacrylamide gel (resolving gel at pH 8.8 only; no SDS was added) was pre-run with Native gel running buffer (25mM Tris and 192mM glycine at pH 8.4; with 1% sodium deoxycholate in the cathodic compartment and without sodium deoxycholate in the anodic compartment) at 40mA for 30 min. The native protein samples were then loaded to the gel and electrophoresed at 25mA for 70 min. When electrophoresis was finished, the protein in the Native gel would be transferred to PVDF membrane (Millipore) via electroblotting.

Electroblotting was conducted in a Semi-dry Hoefer miniVE apparatus (Hoefer SemiPhor, Amersham) at 500mA per gel for 90 min.

## Protein detection

The PVDF membrane was immersed in 5% skim milk in TBST buffer (10mM Tris-HCl, 250mM NaCl, 0.1% Tween-20, pH7.4) at room temperature and gently shook for 1 hour for blocking. After blocking, the PVDF membrane was immersed in primary antibody at concentration instructed by the company in 5% skim milk/TBST for overnight shaking at 4˚C. On the next day, the PVDF membrane was washed with 1× TBST buffer for at least 5 times. Then the membrane was incubated with horseradish peroxidase (HRP)-conjugated secondary antibody at 1:10000 ratio in 5% skim milk/ TBST at room temperature to shake for an hour. After that, the membrane was washed with TBST buffer for at least 5 times. The proteins were then visualized using ECL system (GE Healthcare Life Science).

## Stable cell line selection

HEK293 cells were seeded on 6-well plates (Iwaki) at a concentration of $1 \times 10^5$ per ml and transfected with GeneJuice (Novagen) after 24 hours in a 30μg DNA (BAC) to 3μl GeneJuice ratio. 48–72 hours after transfection, the cells were illuminating fluorescence signals. After ample GFP signals can be observed (~72 hours post-transfection), 100 μg/ml hygromycin B (Thermo Fisher) was added to the culture medium. The culture medium with hygromycin B was changed every 48 hours. After 21 days, the stable cell with BAC incorporated was grown into colonies and ready for single cell picking.

## RNA preparation for RT-PCR

The RNA samples were first treated with DNase I (1 μl Ambion DNase I (Invitrogen), 2 μl Ambion 10× DNase I buffer (Invitrogen), 8 μg RNA and DEPC $H_2O$ for topping up the total volume to 20 μl) for incubation at 37˚C for 30 min followed by 20 min heat inactivation at 65˚C. 10 μl of the treated RNA was added with 3 μl 10 μM primer (random hexamer, oligonucleotides or specific primers) and incubated at 65˚C for 10 min. The solution was then added with 4 μl 5× reverse transcriptase buffer (Roche), 0.5 μl Protector RNase Inhibitor (Roche), 2 μl deoxynucleotide (Roche) and 0.5 μl reverase transcriptase (Roche) for cDNA synthesis. The RT program was 55˚C for 30 min, 85˚C for 5 min and followed by 4˚C. The cDNA samples were then stored at -20˚C.

## qPCR

The transcript levels of IFN-β, IRF3, BPLF1, EBNA1, Zta and Rta were determined by SYBR Premix Ex Taq II (Tli RNase H Plus) (TaKaRa) in StepOne Real-Time PCR System (Thermo Fisher). The transcript levels were normalized with glyceraldehyde 3-phosphate dehydrogenase (GAPDH) transcript level. Sequences of qPCR primers were: IFN-β (5′-F: TTGAATGG GA GGCTTGAAT A; 5′-R: GCCAGGAGGT TCTCAACAAT AG), IRF3 (5′-F: TCTGCCC TCA ACCGCAAAGA AG; 5′-R: TACTGCCTCC ACCATTGGTG TC), BPLF1 (5′-F: ACTG CAAGTT TGGCCGCTTT; 5′-R: ATCTCGTGCC CCTTGAGGAT), EBNA1 (5′-F: GCACC TCCTT CTGTCTGAGC; 5′-R: ACTCTCTGGG CTGCAGAATC), Zta (5′-F: GCACATCTGC TTCAACAGGA; 5′-R: CCAAACATAA ATGCCCCATC), Rta (5′-F: CCTGTCTTGG ACGA GACCAT; 5′-R: AAGGCCTCCT AAGCTCCAAG) and GAPDH (5′-F: AGAAGGCTGG GGCTCATTTG; 5′-R: CTGTGGTCAT GAGTCCTTC).

### EBV primary infection

$1 \times 10^5$ /ml HEK293-M81, HEK293p2089 or HEK293p2089C61A cells were seeded onto 10-cm culture dishes (Thermo Fisher). 4 μg Zta plasmid and 4 μg pCAGGS-gp110 plasmid were transfected to each dish of cells 24 hours after the cells were seeded. Three days post-transfection, the culture medium of 5 dishes was collected and concentrated into 2 ml using Amicon Ultra-15 Centrifugal Filter Unit with Ultracel-100 membrane (Sigma-Aldrich). 300 μl of the concentrated medium was used for infecting each well of the 6-well plate. After 48 hours post infection, the GFP signal was measured by flow cytometry using Flowjo (v.10.0.7).

## Supporting information

**S1 Fig. Induction of full-length BPLF1 suppresses cGAS-STING-, RIG-I N- and TBK1-induced IRF3-Luc activity but not that induced by IRF3-5D.** (A) HEK293M81 cells were transfected with guide-1, guide-2 plasmids or treated with TPA for 48 hours. The total cellular RNA was harvested 48 hours post-transfection. The expression levels of Zta transcripts were measured by RT-qPCR and normalized to those of endogenous GAPDH transcripts. The mean values of three biological replicates (n = 3) were represented by the bars and their respective standard deviations were depicted as the error bars. The statistical significance among selected samples were analyzed using two-tailed Student's t-test for paired samples and the p-values were indicated. (B) HEK293M81 cells were transfected with IRF3-Luc, TK-Luc, increasing doses of guide-2 plasmid, and cGAS + STING, RIG-I N, TBK1 and IRF3-5D expression plasmids for dual luciferase assays. The mean values of three biological replicates (n = 3) were represented by the bars and their respective standard deviations were depicted as the error bars. The statistical significance among selected samples were analyzed using two-tailed Student's t-test for paired samples and the *p* values were indicated. (C) Activity of IκB super-repressor. HEK293 cells were transfected with κB-Luc, which was activated by p65.
(TIF)

**S2 Fig. The DUB activity of BPLF1 mitigated ISD90-induced IRF3-Luc activity and ubiquitination of endogenous STING.** (A) HEK293M81 cells were transfected with IRF3-Luc, TK-Luc, increasing doses of guide-2 plasmid, and cGAS + STING expression plasmids for dual luciferase assays. The cells were transfected with 4 μg/ml ISD90 for 24 hours for analysis of ISD90-induced IRF3-Luc activity. The mean values of three biological replicates (n = 3) were represented by the bars and their respective standard deviations were depicted as the error bars. The statistical significance among selected samples were analyzed using two-tailed Student's t-test for paired samples and the *p* values were indicated. (B) Influence of BPLF1 and C61A on endogenous STING ubiquitination. BPLF1 C61A expression plasmids were transfected into HeLa cells. After 24 hours post transfection, cells were transfected with 4 μg/ml ISD90 for 4 hours. Cells were harvested after 4 hours of ISD90 treatment for co-immunoprecipitation. The endogenous STING molecules were pulled down by anti-STING antibodies. The bound fraction of the immunoprecipitates (IP) and the total lysates (input) were analyzed by Western blotting (WB) with anti-ubiquitin, anti- STING, anti-β-tubulin, anti-cGAS and anti-FLAG antibodies.
(TIF)

**S3 Fig. Reactivation of the HEK293p2089C61A clones by TPA.** NP460, NP460EBV, HEK293p2089 and HEK293p2089C61A cells were seeded on to 12-well plates. TPA is added at 24 hours post-transfection to stimulate lytic transcript production. Cells were harvested in

the next day and RT-qPCR were performed to measure the IFN-β transcript level. GAPDH mRNA was used for normalization.
(TIF)

**S4 Fig. Infection with WT p2089 and mutant p2089C61A viruses.** (A) The DUB activity of BPLF1 promotes EBV infection. BPLF1 and C61A expression plasmids were transfected into HEK293 cells together with cGAS + STING, TBK1 and BPLF1 expression plasmid. After 24 hours post- transfection, cells were infected with 1 m.o.i. of freshly prepared EBV p2089 or p2089C61A. GFP signals were analyzed by flow cytometry 48 hours post-infection. The percentages of GFP$^+$ cells were normalized to the mock infection group. The mean values of two biological replicates (n = 3) were represented by the bars and their respective standard deviations were depicted as the error bars. (B) Total cellular RNA from the infected cells was harvested after flow cytometric analysis and IFN-β transcript expression was measured by RT-qPCR. The expression levels of IFN-β transcript were normalized to those of endogenous GAPDH mRNA. The mean values of three biological replicates (n = 3) were represented by the bars and their respective standard deviations were depicted as the error bars. The statistical significance among selected samples was analyzed using two-tailed Student's t-test for paired samples and the ranges of the $p$ values were indicated ($^*$: $p < 0.05$; and $^{**}$: $p < 0.01$).
(TIF)

## Acknowledgments

We thank Henri-Jacques Delecluse (German Center for Infection Research, Heidelberg, Germany), Lori Frappier (University of Toronto, Toronto, Canada), Wolfgang Hammerschmidt (German Center for Infection Research, Munich, Germany) and Feng Zhang (Broad Institute, Massachusetts, USA) for reagents, and members of Jin laboratory for comments and suggestions on earlier versions of the manuscript.

## Author Contributions

**Conceptualization:** Wai-Yin Lui, Kit-San Yuen, Dong-Yan Jin.

**Data curation:** Wai-Yin Lui, Aradhana Bharti, Nok-Hei Mickey Wong, Sonia Jangra, Michael G. Botelho, Kit-San Yuen, Dong-Yan Jin.

**Formal analysis:** Wai-Yin Lui, Nok-Hei Mickey Wong, Sonia Jangra, Michael G. Botelho, Kit-San Yuen, Dong-Yan Jin.

**Funding acquisition:** Dong-Yan Jin.

**Investigation:** Wai-Yin Lui, Aradhana Bharti, Nok-Hei Mickey Wong, Sonia Jangra, Kit-San Yuen.

**Methodology:** Wai-Yin Lui, Sonia Jangra, Kit-San Yuen.

**Project administration:** Wai-Yin Lui, Kit-San Yuen, Dong-Yan Jin.

**Supervision:** Michael G. Botelho, Kit-San Yuen, Dong-Yan Jin.

**Validation:** Wai-Yin Lui, Kit-San Yuen.

**Visualization:** Wai-Yin Lui, Kit-San Yuen.

**Writing – original draft:** Wai-Yin Lui.

**Writing – review & editing:** Wai-Yin Lui, Aradhana Bharti, Nok-Hei Mickey Wong, Sonia Jangra, Michael G. Botelho, Kit-San Yuen, Dong-Yan Jin.

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
