## [Decision Letter · Decision Letter 0]

20 Sep 2022

Dear Professor Jin,

Thank you very much for submitting your manuscript "Suppression of cGAS- and RIG-I-mediated innate immune signaling by Epstein-Barr virus deubiquitinase BPLF1" for consideration at PLOS Pathogens. As with all papers reviewed by the journal, your manuscript was reviewed by members of the editorial board and by several independent reviewers. In light of the reviews (below this email), we would like to invite the resubmission of a significantly-revised version that takes into account the reviewers' comments.

I am returning your manuscript with three reviews. As you will see, the reviewers raise some important concerns. After reading the reviews and looking at the manuscript, I recommend Major Revision based on the critiques. I am sorry I cannot be more positive at the moment, however we are looking forward to receiving your revision. With a lot of work, the manuscript will be suitable for a resubmission, if you so wish to do so, and further review.

In particular, the reviewers raised concerns about the reliance on gene overexpression in HEK293 cells to induce the signaling pathways studied. There was also strong reviewer concern about the interpretation, and interpretability, of the IRF3 dimerization analysis. Special care would need to be taken to address these issues, as part of a point-by-point response to all reviewer comments.

We cannot make any decision about publication until we have seen the revised manuscript and your response to the reviewers' comments. Your revised manuscript is also likely to be sent to reviewers for further evaluation.

Sincerely,

Christina O'Grady, PhD

Guest Editor

PLOS Pathogens

Erik Flemington

Section Editor

PLOS Pathogens

Kasturi Haldar

Editor-in-Chief

PLOS Pathogens

orcid.org/0000-0001-5065-158X

Michael Malim

Editor-in-Chief

PLOS Pathogens

orcid.org/0000-0002-7699-2064

I am returning your manuscript with three reviews. As you will see, the reviewers raise some important concerns. After reading the reviews and looking at the manuscript, I recommend Major Revision based on the critiques. I am sorry I cannot be more positive at the moment, however we are looking forward to receiving your revision. With a lot of work, the manuscript will be suitable for a resubmission, if you so wish to do so, and further review.

In particular, the reviewers raised concerns about the reliance on gene overexpression in HEK293 cells to induce the signaling pathways studied. There was additional reviewer concern about the interpretation, and interpretability, of the IRF3 dimerization analysis. Special care would need to be taken to address these issues, as part of a point-by-point response to all reviewer comments.

Reviewer's Responses to Questions

**Part I - Summary**

Reviewer #1: The manuscript by Lui et al. investigates the role of the EBV protein BPLF1 in the suppression of cGAS- and RIG-I- mediated innate immune signaling. Using an EBV expression library and an IFNb-luc reporter, the author identified BPLF1 as one of the most potent viral proteins to mitigate IFN-I production. The observed suppression of cGAS-STING- and TBK1-mediated antiviral defense was DUB activity dependent. Furthermore, the authors observed that BPLF1 acts as an effective DUB targeting K63-, K48- and K27-linked ubiquitin moieties on STING and K63- and K48-linked ubiquitin moieties on TBK1. The DUB activity of BPLF1 was also required for its suppression of TBK1-induced IRF3 dimerization. More importantly, in cells carrying the EBV genome with a catalytically inactive BPLF1, the authors observed that EBV virus failed to suppress IFN-b production upon overexpression of cGAS and STING. The data in this story is highly compelling and very nicely done. Overall, this work will be of interest to many investigators interested in EBV biology as well as innate immunity. Below are some suggestions that will help to strengthen the manuscript.

Reviewer #2: Suppression of cGAS- and RIG-I-mediated innate immune signaling by Epstein-Barr virus deubiquitinase BPLF1 by Lui et al is an investigate report on down-regulation of IFN production via the interaction of catalytically active BPLF1. BPLF1 was found to deubiquitinate STING and TBK1 and resulted in suppressive effects on IFN production induced by cGAS-STING, RIG-I and TBK1. Suppressive effects were observed in the presence of functional BPLF1 DUB activity but not when the DUB activity was abolished. The manuscript is well written, and clear. The work is of high significance and provides insight into BPLF1’s ability to evade host innate immune responses. The work is largely convincing but I do have concerns and disagree with conclusions drawn in Figure 7.

Almost all work is done by overexpression. The results would be more relevant if some experiments could be confirmed with endogenous levels of proteins. Have attempts been made to examine protein interactions, ubiquitination state, and expression profiles in the absence of overexpressed TBK1, STING, etc?

Other concerns.

Figure 3F and G. The authors assess the impact of BPLF1 DUB activity on infectivity using HEK293 cells. I assume the authors used 293HEK cells due to ease of transfection but HEK293 cells are not typically infected by EBV. These cells are transfected with STING, TBK1, BPLF1 or C61A and then infected with EBV M81 which contains intact and functional BPLF1. So, in essence C61A or BPLF1 is expressed in addition to endogenous functional levels of BPLF1 if I understand correctly. Additional BPLF1 expression does appear to result in increased infectivity, however it does no appear that additional C61A expression results in a “remarkable decrease” compared to mock as stated in line 235. This should be clarified.

Figure 4. The authors demonstrate that WT BPLF1 removes K48 polyubiquitination from STING which triggers is proteasomal degradation. Deubiquitination results are convincing, however when BPLF1 is present there is not observed increase in STING protein levels. If anything there appears to be a decrease in STING levels. Can the authors discuss this? Could changes in endogenous levels of STING be observed? Additionally in Figure 4B there is a prominent band just above the 34 kDa marker. Does this represent a ubiquitinated form of STING? I would expect a ubiquitinated form of STING to appear around 50KDa. Please clarify.

Figure 5. TBK1 is deubiquitinated by BPLF1 but no increase in protein levels of TBK1 is observed (similar to results with STING in Figure 4). Could ubiquitination of TBK1 be observed without overexpression of TBK1? Both figure 4 and 5 would be more relevant if these could be shown with endogenous levels of STING and TBK1.

Figure 6. IPs demonstrating interaction with BPLF1 and STING are convincing. Again, I would have liked to see this with endogenous levels. Was the opposite IP attempted to demonstrate an interaction between TBK1 and BPLF1?

Figure 7. The authors state that BPLF1 mitigates TBK1-induced dimerization of IRF3 but this is not convincing to me from the data presented. Perhaps I am missing something. In Figure 7A TBK1 clearly induces dimerization of IRF3 as stated in the literature. The addition of WT BPLF1 does reduce the presence of the dimer form but virtually no monomer form is present. I would expect to see an increase in monomer if the dimer is disrupted but his is not observed. The total amount of IRF3 present in the native blot with BPLF1 is much less than observed with C61A or with TBK1 alone and therefore the loss of dimer present may be just a factor of much less total protein. There is a large discrepancy between the denatured amount of IPR3 and the native amount which further complicates the interpretation. Additionally in the absence of TBK1 it looks like BPLF1 induces dimerization (left panels). From the data presented I can’t conclude that BPLF1 disrupts IRF3 dimerization and don’t feel the authors conclusion for Figure 7 is valid. In Figure 7B the authors show an increase in IRF3 mRNA when both IRF3 and BPLF1 are expressed but not when only BPLF1 is expressed (Figure 7C) and state (line 343-344) that the “BPLF1 did not induce IRF3 transcription, but interacted with the CMV promoter on the IFR3 plasmid vector to trigger IRF3 transcriptional activation” yet 7A was performed in the presence of ORF3 over-expression. I don’t think there is convincing data to draw any conclusions from figure 7.

Figure 9 and 10. Construction of the p2089-C61A mutant virus. Local sequencing data and restriction digest patterns help confirm the presence of the intended modification but from personal experience I know this is not always enough to verify the production of the intended mutation. The data in Figure 10 largely falls in line with what one would expect for a valid construct. NGS sequencing, detection of lytic gene products, or creation of a reversion would be helpful to establish that no unintended changes were introduced. The p2089 and p2089-C61A could have been induced and titered on Raji cells to show infectivity by GFP detection. However, in this case, the DUB activity (the key component of this figure) does seem to be maintained and yields convincing results. In figure 10A there is much greater relative expression of C61A compared to WT BPLF1. Can the authors speculate why this may be?

Reviewer #3: The manuscript by Lui et al. describes the inhibition of cytosolic antiviral DNA and RNA sensing pathways, specifically cGAS-STING and RIG-I-MAVS, by the EBV deubiquitinase protein BPLF1. An EBV screen for inhibitors of the cGAS-STING pathway yielded several hits, including BPLF1. Subsequent overexpression as well as induction of endogenous BPLF1 expression by a CRISPRa approach confirmed the inhibitory activity of BPLF1 on the antiviral signaling pathways, which was dependent on its catalytic DUB activity. Further experiments showed that BPLF1 interacted with STING and removed ubiquitin moieties from both STING and TBK1, and also prevented IRF3 dimerization. Finally, introduction of the BPLF1 C61A catalytic mutation in the EBV genome suggested that endogenously expressed BPLF1 suppresses cGAS-STING-mediated IFN induction in infected cells.

The manuscript is overall well written and presents an extensive set of data that includes various in vitro assays complemented with elegant CRISPRa and viral mutagenesis approaches. The data nicely complements the existing literature on evasion of antiviral host sensing pathways by EBV BPLF1 and is interesting to the fields of virology and immunology. However, the manuscript does require the inclusion of essential controls and preferably performing some key assays under more physiological conditions, as detailed below.

**Part II – Major Issues: Key Experiments Required for Acceptance**

Reviewer #1: (No Response)

Reviewer #2: Address Figure 7 conclusions. Perhaps try to show effects of IRF3 dimerization in the absence of PRF3 overexpression or by other methods.

Include or address lack of endogenous studies listed above.

Reviewer #3: General comments:

- The authors have included RIG-I inhibition by BPLF1 as a major point in the manuscript and title. Although RIG-I inhibition is likely due to the observed effects on TBK1 and IRF3 activation, besides a few assays in Figures 1-3, the majority of the assays in the remainder of the paper are focused on cGAS-STING. The authors should consider removing the emphasis on RIG-I, or include additional assays to strengthen this part of the manuscript.

- Although most assays generally seem convincing, they often lack proper positive and/or negative controls or validation of protein expression levels. As indicated for the specific figure panels below, these should be included to confirm the validity of the assays.

- The authors rely almost entirely on overexpression of signaling components in HEK293T cells for activation of the antiviral pathways. Performing a few key experiments in more physiological settings, for example by using exogenous stimuli or virus infection, would significantly strengthen the manuscript. Furthermore, since the authors have prepared a BPLF1 C61A mutant virus, they could also perform infection experiments in wildtype and cGAS-deficient cells to determine the relevance of BPLF1-mediated cGAS-STING evasion during authentic infection.

Specific comments:

Figure 1:

- The authors show two different doses of cotransfected BPLF1 in panels B-D. However, both doses are equally potent in suppressing IFNB-Luc. The authors could consider including a lower dose of BPLF1 to show a dose-dependent effect.

Figure 2:

- The induction of luciferase activity by IRF3 in panels B and C is rather low (2 to 4-fold). The authors should comment on why this is the case and if that affects their ability to observe BPLF1-mediated inhibition of IFNB-luciferase induction. These assays would also become more reliable if Western blot analyses verifying BPLF1 levels were included.

- In panel G, it cannot be ruled out that the CRISPRa approach leads to general EBV reactivation and lytic gene expression. To validate the specificity of their CRISPRa approach in upregulating only BPLF1, the authors need to compare the abundance of one or more other lytic EBV genes besides BPLF1 between the mock and guide-1/2 conditions.

- In panel H, the authors should include a control that is not affected by BPLF1 expression (e.g. IRF3 overexpression) to verify that sgRNA-cotransfection does not result in a non-specific reduction in IFNB-Luc activity.

- In panel J, the authors conclude that the effects of NF-κB are not involved in their assay since they do not observe an inhibitory effect by IκB-sr. However, the authors need to show that they achieve functional IκB-sr expression, for example by including an NF-κB luciferase reporter whose induction is expected to be inhibited by IkB-sr.

Figure 3:

- Since the differences in GFP expression in panel F are relatively small, it is hard to quantitatively assess these results. The authors should include a quantification of the percentage of GFP+ (infected) cells, for example by flow cytometry. Especially since the difference in EBV DNA content in panel G is also relatively minor (1.5 fold increase). Furthermore, to allow appreciation of the effect of cGAS/STING and TBK1 overexpression on virus infection efficiency, the authors should also include a sample without overexpression of these signaling molecules.

Figure 4:

- In panel A, the authors should include a Western blot showing successful and equal precipitation of FLAG-STING in each of the samples (IP: FLAG, WB: FLAG).

- For panels B and C, the images are partly overexposed and cannot be interpreted. The authors need to (additionally) include lower exposures of these blots.

Figure 5:

- Some images cannot be interpreted due to overexposure. The authors need to include (additional) lower exposures.

Figure 7:

- The authors show that BPLF1 expression enhances ectopic IRF3 expression, which they conclude is mediated by the CMV promoter. Can the authors comment on whether they have additional evidence for the involvement of the CMV promoter, and whether this effect affects other assays presented in the manuscript, e.g. the luciferase assays?

Figure 8:

- In panel C, the authors compare BPLF1 and LMP1 expression and conclude that AGS-BX1 cells express relatively high levels of BPLF1. Since LMP1 is a latent transcript, the relative increased expression of BPLF1 could mean that there is more BPLF1 in latently infected cells, or that a higher percentage of AGS-BX1 cells is undergoing (spontaneous) lytic reactivation in comparison to the other cell lines. The authors should comment on this.

- In panel F, the authors should consider including a non-targeting control sgRNA to show specificity of the observed inhibition of IFN mRNA induction by the BPLF1-targeting sgRNAs.

Figure 10:

- As acknowledged by the authors in the Discussion section, clonal differences between the p2089 and p2089-C61A variants may cause differences in the presented assays. Even though the authors have tried to diminish these effects by including several independent clones, a control should be included (e.g. IRF3 transfection) that is not affected by BPLF1 to verify that the signaling pathway leading to IFNB-Luc activation is similarly active in each set of clones.

- Moreover, only a minor proportion of the cells is likely undergoing spontaneous lytic EBV infection and expresses BPLF1. The cGAS-STING pathway is thus expected to be induced primarily in cells not expressing BPLF1. The authors should comment on why they still observe a robust reduction in IFNB-Luc activity in the p2089 samples.

**Part III – Minor Issues: Editorial and Data Presentation Modifications**

Reviewer #1: The suppression of IFN-b production by BPLF1 is tested in artificial conditions such as overexpression of cGAS+STING, RIG-I-N and TBK1. It would be nice to test the ability of BPLF1 to suppress signaling in response to a ligand, such as ISD90, triphosphorylated RNA, or polyI:C.

In Fig3F, please add a control with no cGAS+STING expression. Also, the difference of the percentage of GFP positive cells is minor. Quantification by flow cytometry would be ideal

In Fig4 and 5, please add blot for WT and C61A BPLF1.

In Fig4A, WT BPLF1 expression clearly reduced F-STING expression comparing lane 2 and 4. However, the author claimed that “The steady-state level of STING remained constant.” Although I agree that the perturbing the PTM of STING is more than likely to be the major effect of BPLF1, the author should be more careful about how to describe the data.

Fig6C, how does BPLF1 perturb TBK1 PTM if it does not interact with TBK1? It is important to prove the interaction between TBK1 and BPLF1. Crosslinking the cell with DSG or DSP will be beneficial to capture transit protein-protein interactions.

Line 342, the authors showed no evidence to support BPLF1 “interacted with the CMV promoter on the IRF3 plasmid vector to trigger IRF3 transcriptional activation.” Please modify the text.

In Fig9D, it seems like clone1 and clone 2 have extra bands at the top comparing to p2089. Please show uncropped figure.

Reviewer #2: Other issues mentioned above.

Reviewer #3: (No Response)

PLOS authors have the option to publish the peer review history of their article (what does this mean?). If published, this will include your full peer review and any attached files.

Reviewer #1: No

Reviewer #2: No

Reviewer #3: No
---

## [Decision Letter · Decision Letter 1]

17 Jan 2023

Dear Professor Jin,

Thank you very much for submitting your manuscript "Suppression of cGAS- and RIG-I-mediated innate immune signaling by Epstein-Barr virus deubiquitinase BPLF1" for consideration at PLOS Pathogens. As with all papers reviewed by the journal, your manuscript was reviewed by members of the editorial board and by several independent reviewers. The reviewers appreciated the attention to an important topic. Based on the reviews, we are likely to accept this manuscript for publication, providing that you modify the manuscript according to the review recommendations.

The reviewers were largely satisfied with the revised submission, but Reviewer #2 is concerned about the description of Zta and BMRF2 mRNA levels as "similar" between the C61 mutant and the wild-type. This reviewer requests that you update the text to address the difference in mRNA levels seen in Figure S3 (much as you have updated the text to address a similar issue in Figure 10A).

Sincerely,

Christina O'Grady, PhD

Guest Editor

PLOS Pathogens

Erik Flemington

Section Editor

PLOS Pathogens

Kasturi Haldar

Editor-in-Chief

PLOS Pathogens

orcid.org/0000-0001-5065-158X

Michael Malim

Editor-in-Chief

PLOS Pathogens

orcid.org/0000-0002-7699-2064

Reviewer Comments (if any, and for reference):

Reviewer's Responses to Questions

**Part I - Summary**

Reviewer #1: The authors have satisfactorily addressed my concerns.

Reviewer #2: This is a revision from a previous submission. I am largely satisfied with the responses to my review. The authors performed experiments with endogenous levels of STING and demonstrated ubiquitination but they did not do this with TBK1. The authors have adjusted the conclusion for Figure 7.

In point 7 of the rebuttal the authors show mRNA levels for ZTA, RTA, and BMRF2 (S3) and in their response state the levels are "similar." However ZTA mRNA levels look to be much higher with the C61 mutants. They are also somewhat higher with the BMRF2 mRNA. In Figure 10A there is a much greater relative expression of C61 compared to WT and the authors speculate this may be due changes occurring in culture over time and suggest differences in genome methylation. The unknown difference that may have evolved during passage in culture could also contribute to the observed phenotype. The difference in mRNA levels of ZTA and BMFRF2 should be mentioned in the text.

Reviewer #3: The authors have significantly strengthened the manuscript by performing the requested assays and including additional controls. The concerns raised previously have been adequately addressed.

**Part II – Major Issues: Key Experiments Required for Acceptance**

Reviewer #1: (No Response)

Reviewer #2: (No Response)

Reviewer #3: (No Response)

**Part III – Minor Issues: Editorial and Data Presentation Modifications**

Reviewer #1: (No Response)

Reviewer #2: The authors mention the difference BPLF1 levels for the C61 mutations and provide a possible reason in the discussion (Figure 10A). Differences in MRNA levels of Zta, and BMRF2 should also be mentioned (S3).

Reviewer #3: (No Response)

PLOS authors have the option to publish the peer review history of their article (what does this mean?). If published, this will include your full peer review and any attached files.

Reviewer #1: No

Reviewer #2: No

Reviewer #3: No

Figure Files:

Data Requirements:

Reproducibility:

References:

---

## [Editor Report · Decision Letter 2]

6 Feb 2023

Dear Professor Jin,

We are pleased to inform you that your manuscript 'Suppression of cGAS- and RIG-I-mediated innate immune signaling by Epstein-Barr virus deubiquitinase BPLF1' has been provisionally accepted for publication in PLOS Pathogens.

Best regards,

Christina O'Grady, PhD

Guest Editor

PLOS Pathogens

Erik Flemington

Section Editor

PLOS Pathogens

Kasturi Haldar

Editor-in-Chief

PLOS Pathogens

orcid.org/0000-0001-5065-158X

Michael Malim

Editor-in-Chief

PLOS Pathogens

orcid.org/0000-0002-7699-2064
---

## [Editor Report · Acceptance letter]

15 Feb 2023

Dear Professor Jin,

We are delighted to inform you that your manuscript, "Suppression of cGAS- and RIG-I-mediated innate immune signaling by Epstein-Barr virus deubiquitinase BPLF1," has been formally accepted for publication in PLOS Pathogens.

Best regards,

Kasturi Haldar

Editor-in-Chief

PLOS Pathogens

orcid.org/0000-0001-5065-158X

Michael Malim

Editor-in-Chief

PLOS Pathogens

orcid.org/0000-0002-7699-2064